# PAROAttention: Pattern-Aware ReOrdering for Efficient Sparse and Quantized Attention in Visual Generation Models

**Tianchen Zhao**[*1,2], **Ke Hong**[*1], **Xinhao Yang**[*1], **Xuefeng Xiao**[2], **Huixia Li**[2], **Feng Ling**[2], **Ruiqi Xie**[1], **Siqi Chen**[1], **Hongyu Zhu**[1], **Zhang Yichong**[1], **Yu Wang**[†1]

[1]Tsinghua University, [2]ByteDance Seed

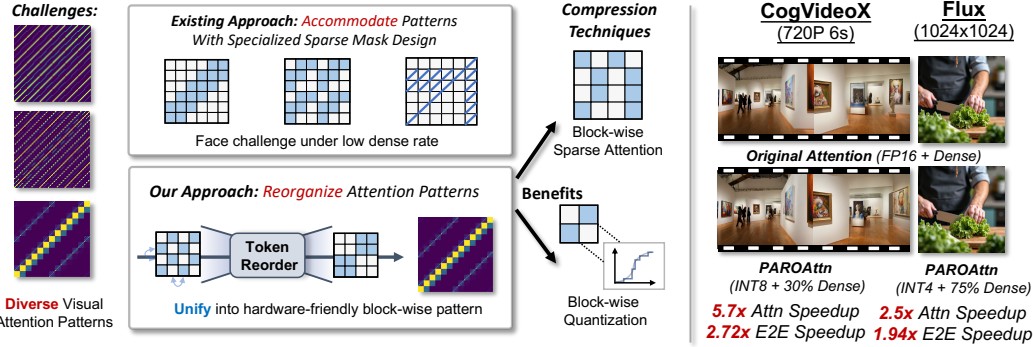

Figure 1: **PAROAttention** unifies the diverse attention patterns through token reorder, which benefits both the sparsification and quantization. It achieves nearly identical generation result from full-precision baseline without metrics degradation, under lower density (**20%-30%**) and bitwidth (**INT8/INT4**), achieving a **1.9~2.7×** end-to-end latency speedup.

## Abstract

In visual generation, the quadratic complexity of attention mechanisms results in high memory and computational costs, especially for longer token sequences required in high-resolution image or multi-frame video generation. To address this, prior research has explored techniques such as sparsification and quantization. However, these techniques face significant challenges under low density and reduced bitwidths. Through systematic analysis, we identify that the core difficulty stems from the dispersed and irregular characteristics of visual attention patterns. Therefore, instead of introducing specialized sparsification and quantization design to accommodate such patterns, we propose an alternative strategy: "*reorganizing*" the attention pattern to alleviate the challenges. Inspired by the local aggregatin nature of visual feature extraction, we design a novel **Pattern-Aware token Re-Ordering (PARO)** technique, which unifies the diverse attention patterns into a hardware-friendly block-wise pattern. This unification substantially simplifies and enhances both sparsification and quantization. We evaluate the performance-efficiency trade-offs of various design choices and finalize a methodology tailored for the unified pattern. Our approach, **PAROAttention**, achieves video and image generation with lossless metrics, and nearly identical results from full-precision (FP) baselines, while operating at notably lower density (**20%-30%**) and bitwidth (**INT8/INT4**), achieving a **1.9~2.7×** end-to-end latency speedup.

---

[*]Equal contribution. work done when Tianchen Zhao intern at Bytedance
[†]Corresponding author: Yu Wang (yu-wang@tsinghua.edu.cn).

39th Conference on Neural Information Processing Systems (NeurIPS 2025).

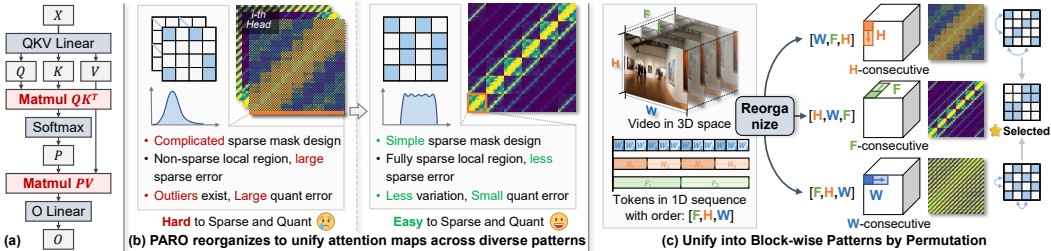

Figure 2: **The Motivation of PAROAttention.** (a) The computational flow of transformer. (b) The challenge for sparsification and quantization caused by visual attention pattern, and how PAROAttention addresses it. (c) The illustration of 3D feature, and 1D token sequence with different orders.

## 1 Introduction

Diffusion transformers (DiTs [35]) have garnered significant research interest in visual generation tasks. However, their excessive resource cost poses challenges for broader applications. The adoption of "3D full attention" in models like CogVideoX [56] further increases token length. The quadratic complexity of attention mechanisms in token length results in substantial memory consumption and computational overhead when processing such long sequences. For instance, generating a 49-frame 6-second 720P video involves 17K token length[2]. Attention computation contributes to the majority of the total latency, making it the primary bottleneck and requiring urgent optimization. As shown in Fig. 2(a), the two matrix multiplications $QK^T$ and $PV$ ($P$ is the attention map after softmax) has a computation cost quadratic to token length, making them the primary bottleneck in attention.

Previous research has explored sparse attention mechanisms [58, 55] and quantization [61, 59, 60] to accelerate attention. While these techniques achieve notable success in language models [3, 8, 20, 18], they cannot be directly applied to visual generative models due to **distinct attention patterns**. As shown in Fig. 2-(b), **different attention heads exhibit diverse patterns**. These patterns vary not only in type (e.g., blockwise, multi-diagonal) but also in their structural characteristics—such as the number, width, and spacing of the diagonals. Recent sparse attention methods [57, 64, 45] explore designing specialized sparse masks for visual models, but still face significant challenges in maintaining quality at lower density rates (< 50%). Quantization techniques tailored to visual generation remain underexplored. Existing methods struggle to efficiently quantize $PV$ computations to lower-bit integers (e.g., INT8/INT4), and often remain at FP16/FP8.

We conduct a systematic analysis of the underlying reason for suboptimal performance of existing techniques under low density and bitwidths (as discussed in Sec. 3). Our findings reveal that the core challenge of sparsification and quantization stem from distinct characteristics of visual attention patterns. For sparsification, the dispersed and dynamically changing attention patterns lead to the absence of locally sparse regions, resulting in sparsification error. For quantization, the presence of multiple "diagonal" values act as outliers within data group, thereby increasing quantization error. Different from existing methods that design specialized sparsification and quantization techniques to accommodate the diverse patterns, we propose an alternative direction: **To "reorganize" the attention patterns to ease the difficulty the design of both sparsification and quantization methodologies.**

To design proper technique to "reorganize" the attention pattern, we further analyze the underlying causes of the diverse visual attention patterns. Prior literature [33] reveals the local aggregation nature of visual attention, which suggest that attention tends to capture relationships between neighboring pixels. In vision transformers, the 3D physical space are flattened into 1D token sequences, which disrupts the data adjacency. For example, as shown in Fig. 2-(c), the neighboring tokens along $F$-dimension in 3D space are not consecutive, and have the same interval of $H \times W$ in 1D token sequence of default order $[F, H, W]$. The aggregation of these tokens with equal intervals forms the "multi-diagonal" pattern. Therefore, the multi-diagonal and block-wise patterns are intrinsically the same, representing local aggregation in different dimensions. By applying token reordering, we rearrange the tensor layout (e.g., permute from $[F, H, W]$ to $[H, W, F]$)) for each head to keep

---

[1]All videos in the figure are provided in the supplementary.

[2]We use this setting as example for most of the description below, the image token length $N = 17550 = F * W * H = 13 * 30 * 45$, where $F, H, W$ stands for the frame number, height, and width in the latent space.

elements of the local aggregation dimension contiguous. **The "Pattern-Aware token ReOrder (PARO)" could transform diverse patterns into a unified, hardware-friendly block-wise pattern.**

Furthermore, we analyze the trade-offs among design choices in terms of accuracy, and efficiency, and design specialized sparsification and quantization techniques tailored to the unified block-wise pattern, constructing the **PAROAttention** method. We summarize our contributions as follows:

- We analyze and identify the key challenges of attention sparsification and quantization as unique attention pattern characteristic, and propose to address it from a novel direction of token reorder to reorganize and unify the attention pattern.
- We compare the strength and weakness of design choices, and develop specialized methodologies tailored for the unified pattern, along with CUDA kernels for practical acceleration.
- PAROAttention, achieves generation with lossless metrics, and nearly identical results from full-precision (FP) baselines, while operating at notably lower density (**20%-30%**) and bitwidth (**INT8/INT4**), achieving a **1.9∼2.7×** end-to-end latency speedup.

## 2 Related Works

### 2.1 Visual Generative Models

Diffusion transformers [35], which leverage transformer architectures [40, 9], have achieved outstanding performance and are widely adopted by recent image generation models such as PixArt [6] and Flux [19]. For video generation, earlier approaches, such as OpenSORA [15], apply "spatial-temporal" attention, which performs attention separately along the spatial and temporal dimensions. Other recent models like CogVideoX [56], Wan [41] adopt "3D full attention" instead, processing all spatial tokens across all frames simultaneously. The increased model size and token length pose significant challenges for efficient deployment.

### 2.2 Sparse Attention for Generative Models

Existing sparse attention research [3, 8, 5, 48, 29, 65] primarily focuses on designing sparse masks aligned with specific attention patterns, such as sliding window patterns [67, 49, 50] and attention sink patterns [51, 11] commonly observed in language models. However, visual generation involves unique attention patterns that require new forms of specialized sparse mask design. Recent studies have explored various mask designs tailored for visual generation, including window-based approaches (e.g., DiTFastAttn [57], STA [64]), spatial temporal patterns (e.g., SparseVideoGen [46, 54]), and hybrid mask combinations (e.g., MInference [18]). In contrast, SpargeAttention [62] does not rely on predefined sparse masks but instead generates them online based on $QK$ embeddings. Despite these advancements, the dispersed and diverse distribution of attention values in visual tasks prevents these mask patterns from maintaining quality under low density. In this work, we address this challenge by reorganizing attention patterns through token reordering.

### 2.3 Quantization for Generative Models

Quantization [37, 66, 22, 31, 23] has proven to be highly effective across a wide range of applications. For visual generation, prior research [69, 36, 24, 70, 21, 16, 26] has identified unique challenges associated with quantizing DiTs, they primarily focused on quantizing the linear layers in transformers. However, the increasing token length has shifted the bottleneck to the attention mechanism. Recent advancements [61, 59], have explored quantizing $QK^T$ to INT4/8 while employing FP8 (8 bit floating-point) for $PV$. In this work, we address the unique challenges of attention quantization in visual generation, analyzing the underlying reasons behind the difficulty of quantizing the $P$ matrix. We further extend our investigation to explore INT4/8 quantization for the $PV$ computation.

## 3 Preliminary Analysis

**Key Challenge for Sparse Attention:** The goals of sparse attention design are two-folded: (1) preserving *algorithmic performance* by avoiding the removal of important attention values, and (2) enabling practical *hardware acceleration*. Since arbitrary sparsity patterns could not achieve practical

acceleration [7], proper structured sparsity is necessary. However, as illustrated in Fig. 2, the dispersed and diverse nature of visual attention patterns presents significant challenges for designing structured sparsity. Firstly, diverse attention patterns vary in type (block-wise, multi-diagonal, and diagonal-in-block), as well as in structural characteristics (the number, width, and spacing of diagonals). These patterns also change dynamically across different timesteps and prompts. Designing proper structured sparsity pattern that accommodates all these variations and generalizes across different scenarios is extremely difficult. Secondly, the dispersed attention distribution makes it challenging to form fully sparse regions, thereby inevitably introducing errors in structured sparsity. Given these challenges, **improving sparse mask design to accommodate diverse patterns may have limited effectiveness. Instead, we propose an alternative direction:** *reorganizing* the attention pattern.

**Key Challenge for Attention Quantization:** The goal of quantization design is to minimize quantization error, and we begin by analyzing its sources. As shown in prior work [69], a major source of quantization error arises from large variations within a data group. In such cases, the computed scaling factor becomes excessively large, compressing the majority of values toward zero and leading to significant quantization error. Upon analysis, we find that the distributions of $Q$, $K$, and $V$ do not exhibit substantial variation. The primary challenge lies in properly quantizing the attention matrix $P$. As illustrated in Fig. 2, in the "diagonal-like" patterns, the larger values along the diagonals act as "outliers" within each local region (i.e., quantization group), leading to substantial rounding errors. Addressing this issue is crucial for reducing quantization error. **This also points to the need for** *reorganizing* **the attention distribution to reduce outliers.**

## 4 Methodology

### 4.1 Pattern-aware Token Reordering (PARO)

As concluded in Sec. 3, it is crucial to introduce a technique that reorganizes the attention distribution to mitigate challenging characteristics and promote a structure more favorable to sparsification and quantization. To explore this, we investigate the root causes of the diverse attention patterns. Inspired by the local aggregation nature discussed in Sec. 1, we observe that diverse patterns actually represents local aggregation along different dimensions. They could potentially be transformed into a localized block-wise pattern by gathering the locally aggregated tokens together through token reordering.

However, determining the optimal reorder strategy for each attention head is a non-trivial challenge. First, the large number of tokens ($N_{\text{token}}$) leads to a vast search space of possible reorders, making optimization difficult. Second, the reorder strategy must be carefully designed to minimize hardware overhead. Third, it should simultaneously satisfy the distinct requirements of sparsification and quantization: sparsification benefits from fully sparse local regions, whereas quantization requires balanced distributions within local regions. To address these challenges, we focus on a specific subset of reorder - ***"permutations"***, inspired by the observation that certain attention heads tend to aggregate information locally along a particular dimension. In the case of 3D video generation, where tokens are structured along three dimensions $[F, H, W]$, we limit the reordering space to the six possible permutations ($P_3^3 = 6$). We verify that the optimal permutation is sufficient to produce unified, block-wise patterns through empirical analysis as seen in Fig. 3. We further elaborate on how we address the above mentioned challenges as follows:

**Minimize Hardware Overhead:** We first verify that the optimal permutation remains consistent across different timesteps and prompts. Based on this observation, we determine the permutation order offline, eliminating the need for runtime overhead associated with generating the reordering strategy. Consequently, the only remaining overhead is the online application of the permutation itself, which primarily involves data movement. This cost can be significantly reduced by fusing the permutation operation with preceding kernels. After fusion, explicit data movement from global memory on the GPU is avoided—only the output write-back addresses need to be adjusted. The resulting overhead is less than 1% of the preceding kernel (e.g., LayerNorm), and thus negligible compared to the cost of attention computation.

**Metric for Permutation Order:** As discussed above, sparsification and quantization prefer different distributions. We design separate metrics and combine them to to determine the permutation order for each head. Given the post-softmax attention map $P \in \mathbb{R}^{N \times N}$, $N = k \times b$, where $b$ is the block size. $P$ is tiled into $k \times k$ sub-matrices $P_{ij} \in \mathbb{R}^{b \times b}$. For sparsification, we choose a relatively small value $\epsilon$ (e.g., 1e-3), and classify the block as sparse if the vast majority of values (over threshold $\sigma$, e.g.,

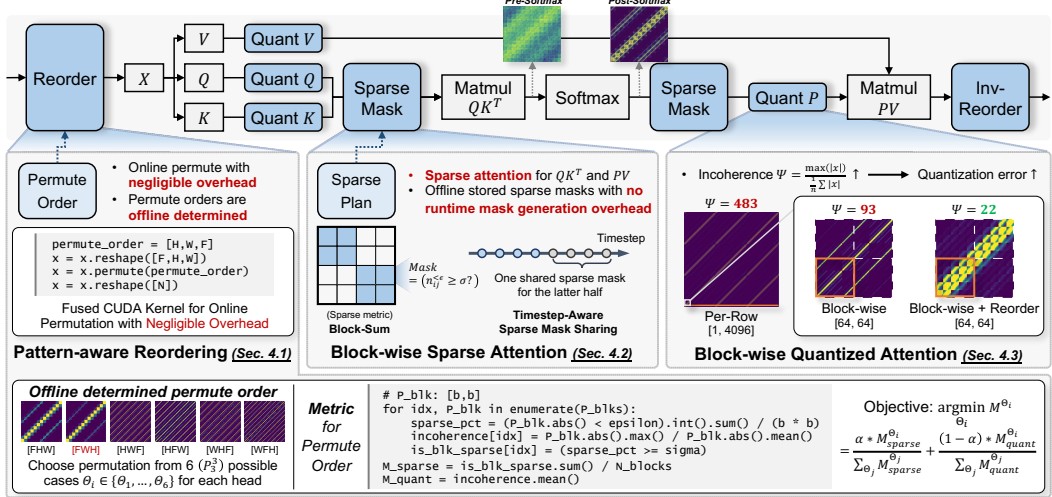

Figure 3: **The overall framework of PAROAttention.** The pattern-aware token reordering (PARO) is applied to unify the attention pattern into hardware-friendly block pattern. Sparse attention and quantization techniques are designed tailored for this pattern.

90%)) within the block are smaller than $\epsilon$. The percentage of such sparse blocks is adopted as the sparse metric $M_{sparse}$, it could be calculated as follows, where $\mathbb{I}$ stands for the indicator function.

$$n_{ij}^{<\epsilon} = \sum_{m=1}^{b}\sum_{n=1}^{b}\mathbb{I}\left(|P_{ij}(m,n)| < \epsilon\right), M_{sparse} = \frac{1}{k\times k}\sum_{i=1}^{k}\sum_{j=1}^{k}\mathbb{I}\left(\frac{n_{ij}^{<\epsilon}}{b\times b} \geq \sigma\right). \quad (1)$$

For quantization, we follow the previous methods [69] and adopt "incoherence" $\Psi$ (maximum value divided by the mean absolute value) as an indicator of quantization difficulty within data group $x \in \mathbb{R}^g$. The quantization metric $M_{quant}$ is defined as follows:

$$\Psi(x) = \frac{\max(|x|)}{\frac{1}{g}\sum|x|}, \quad M_{quant} = \frac{1}{k\times k}\sum_{i=1}^{k}\sum_{j=1}^{k}\Psi(P_{ij}). \quad (2)$$

Finally, the $M_{sparse}$ and $M_{quant}$ are normalized across all possible permutations $\Theta_i \in \{\Theta_1, ..., \Theta_6\}$, and combined as the final metric $M$. The weighting coefficient $\alpha$ controls the relative importance of the two aspects, and the permutation with the lowest $M^{\Theta_i}$ is chosen.

$$M^{\Theta_i} = \alpha * \frac{M_{sparse}^{\Theta_i}}{\sum_{\Theta_j} M_{sparse}^{\Theta_j}} + (1 - \alpha) * \frac{M_{quant}^{\Theta_i}}{\sum_{\Theta_j} M_{quant}^{\Theta_j}}. \quad (3)$$

## 4.2 Block-wise Sparse Attention

After applying permutation, we obtain attention maps with unified and regular block-wise patterns. We further elaborate on comparing the strengths and limitations of different design choices to conclude the final PAROAttention sparsification design.

**Static vs. Dynamic Sparse Attention:** There are two major schemes for sparse attention: the dynamic approach, which predicts sparse masks online, and the static approach, which calibrates sparse masks offline. Since the PARO attention reorganization is compatible with both schemes, we summarize their respective strengths and limitations below and justify our final selection:

(1) In terms of preserving algorithmic performance, the dynamic approach bases on $QK$ embeddings to predict the sparse mask. However, as shown in Fig. 3, the pre-softmax attention map ($QK^T$) contains relatively uniform values and lacks distinguishable sparse patterns, making accurate prediction of the attention pattern difficult. Furthermore, the $QK$ embeddings often need to be downsampled to reduce computational overhead, which further compromises prediction accuracy. The static approach, on the other hand, benefits from access to the more informative post-softmax attention patterns. Nevertheless, it still faces challenges in designing sparse masks that accommodate the highly diverse, and dynamically changing attention distributions. Fortunately, with our token reordering strategy that transforms attention into a unified block-wise structure, these challenges are significantly alleviated.

(2) In terms of hardware efficiency, the dynamic approach introduces runtime overhead for online sparse mask prediction. Reducing this cost often comes at the expense of prediction accuracy. The static approach, while avoiding this cost, incurs memory overhead for storing sparse masks and offline calibration overhead. To address this, we develop techniques to minimize these costs.

**Minimize Hardware Overhead:** We ensure that the sparsification design of PAROAttention introduces minimal overhead by incorporating the following techniques:

(1) Timestep-Aware Sparse Mask Sharing: As discussed above, offline determined static sparse mask face challenges when generalizing to dynamically changing sparse patterns. To address this, we systematically analyze the similarity of attention maps across multiple dimensions. Since sparsification is applied only to image tokens (account for 99% of all tokens), we observe extremely high similarity across different prompts (cosine similarity $\geq 0.99$). However, lower similarity is witnessed across the timestep dimension and we adopt timestep-wise sparse masks. Despite improving accuracy, timestep-wise sparse masks increase memory usage. We observe that attention patterns change most during the early timesteps. Thus, we apply distinct timestep-wise masks only for the first half of timesteps and reuse a common mask for the remainer. We further reduce runtime memory cost by prefetching sparse masks for each head during inference, eliminating the need to store all masks simultaneously. When stored as binary bitmasks, each head's sparse mask requires only 9.2 KB, making the overall storage and data movement overhead negligible.

(2) Block-Aligned Sparse Granularity for Efficient CUDA Execution: FlashAttention processes attention in a block-wise manner, we align our sparsification granularity with the FlashAttention block size. This allows for an extremely simple CUDA implementation, where entire blocks can be skipped without additional branching or logic overhead.

(3) Efficient Offline Mask Generation: Although static methods allow sophisticated sparse metric design, we find that a simple block sum thresholding is sufficient. It also enables explicit control over density. Due to strong generalization across prompts, only 1–2 prompts are needed to determine the permutation order and generate the sparse masks, which only involves minute-level cost.

## 4.3 Block-wise Quantized Attention

As discussed in Sec. 3, the major source of quantization error comes from the data variation within quantization group. Prior literature [4, 69] introduces a metric a metric "incoherence" $\Psi$, as shown in eq. (2), to measure the relative "sharpness" of the data distribution within a group. Data groups with higher incoherence are more challenging to quantize. We conclude and justify our design choice for PAROAttention quantization design to reduce incoherence and ensure hardware efficiency as follows: **(1) Block-Aligned Quantization Granularity (Grouping)**: We emphasize the importance of aligning the quantization grouping with the FlashAttention block size, considering both algorithmic performance and hardware efficiency. As illustrated in Fig. 3, adopting a naive per-row quantization scheme (analogous to per-token grouping for $Q$ and $K$) is not only incompatible with the block-wise processing paradigm of FlashAttention but also introduces substantial incoherence due to the inherently "diagonal" structure of visual attention patterns. This highlights the necessity of block-wise quantization grouping. However, even within local blocks, the diagonal distribution persists, leading to high incoherence (e.g., $\bar{\Psi} = 93$), which necessitates further optimization to reduce quantization error. **(2) Token Reorder for Incoherence Reduction:** Prior work has proposed distribution-balancing quantization techniques for linear layers, such as scaling [27] and rotation [4]. However, these approaches are not applicable to the $PV$ computation in FlashAttention due to the iterative update mechanism of $P$, which is not explicitly materialized in order to save memory. Instead, we explore a novel direction: tuning the attention distribution via token reordering, which groups similar attention values together. As shown in Fig. 3, this reordering significantly reduces incoherence, thereby effectively mitigating quantization error.

# 5 Experiments

## 5.1 Experimental Setup

**Video and Image Generation:** For video generation, we apply PAROAttn to the CogVideoX-5B [56] and Wan [41] (see Appendix Sec.1) model for 720P 6/10-second video with 30 sampling steps. To verify generalization, we collect calibration data with the first 2 prompts of the CogVideo example

Table 1: **Performance of PAROAttention CogVideoX text-to-video generation on VBench prompts.** Baselines are evaluated using their official codebases. For fair comparison, we configure SparseVideoGen without skipping sparsification during the first 30% of timesteps. The "SpargeAttn (0.5 + PARO)" denotes the SpargeAttention method augmented with token reordering (PARO).

| Type | Method | Efficiency Dense Rate / Bitwidth | Video Quality Metrics CLIPSIM↑ | VQA↑ | ΔFScore↓ | FP Diff. Metrics FVD-FP16↓ | PSNR↑ | SSIM↑ | CosSim↑ |
|---|---|---|---|---|---|---|---|---|---|
| - | FP16 Full Attn. | 100.0% | 0.203 | 92.53 | 0.000 | 0.000 | ∞ | 1.000 | 1.000 |
| Sparse | DiTFastAttn (0.5) | 50.0% | 0.197 | 90.43 | 0.740 | 0.904 | 15.40 | 0.603 | 0.920 |
| | MInference (0.5) | 50.0% | 0.197 | 86.02 | 2.250 | 0.368 | 16.54 | 0.696 | 0.945 |
| | SpargeAttn (0.5) | 50.0% | 0.198 | 87.72 | 1.154 | 0.347 | 16.80 | 0.683 | 0.938 |
| | SpargeAttn (0.5 + PARO) | 50.0% | 0.198 | 89.26 | 0.671 | 0.259 | 17.32 | 0.693 | 0.948 |
| | SparseVideoGen (0.5) | 50.0% | 0.198 | 90.14 | 0.568 | 0.186 | 18.50 | 0.755 | 0.960 |
| | PAROAttn (0.5) | 50.0% | 0.203 | 92.56 | 0.103 | 0.068 | 29.14 | 0.936 | 0.997 |
| | SpargeAttn (0.3) | 30.0% | 0.197 | 86.74 | 1.231 | 0.375 | 15.22 | 0.642 | 0.912 |
| | SpargeAttn (0.3 + PARO) | 30.0% | 0.197 | 89.96 | 1.142 | 0.339 | 16.89 | 0.683 | 0.946 |
| | SparseVideoGen (0.3) | 30.0% | 0.197 | 89.54 | 0.589 | 0.241 | 17.73 | 0.725 | 0.954 |
| | PAROAttn (0.3) | 30.0% | 0.204 | 92.66 | 0.101 | 0.153 | 22.89 | 0.829 | 0.984 |
| | PAROAttn (0.2) | 20.0% | 0.203 | 92.42 | 0.151 | 0.151 | 19.39 | 0.744 | 0.962 |
| | PAROAttn (0.125) | 12.5% | 0.201 | 90.21 | 0.203 | 0.218 | 15.93 | 0.690 | 0.937 |
| Quant | RTN (INT8) | QK (INT8), PV (INT8) | 0.190 | 92.09 | 0.571 | 0.480 | 18.88 | 0.750 | 0.956 |
| | RTN (INT4) | QK (INT4), PV (INT4) | 0.184 | 69.25 | 3.360 | 1.446 | 11.99 | 0.500 | 0.905 |
| | SageAttn | QK (INT8), PV (FP16) | 0.203 | 92.24 | 0.131 | 0.047 | 29.58 | 0.927 | 0.997 |
| | SageAttnV2 | QK (INT4), PV (FP8) | 0.200 | 88.79 | 2.460 | 1.750 | 24.46 | 0.824 | 0.979 |
| | PAROAttn (INT8) | QK (INT8), PV (INT8) | 0.203 | 92.57 | 0.096 | 0.026 | 29.01 | 0.935 | 0.996 |
| | PAROAttn (INT4) | QK (INT4), PV (INT4) | 0.200 | 89.24 | 0.876 | 1.382 | 24.16 | 0.822 | 0.985 |
| Sparse +Quant | PAROAttn (0.3+INT8) | 30% + QK, PV (INT8) | 0.201 | 91.68 | 0.884 | 0.533 | 21.49 | 0.779 | 0.976 |
| | PAROAttn (0.5+INT4) | 50% + QK, PV (INT4) | 0.200 | 90.42 | 0.967 | 1.431 | 24.34 | 0.827 | 0.986 |

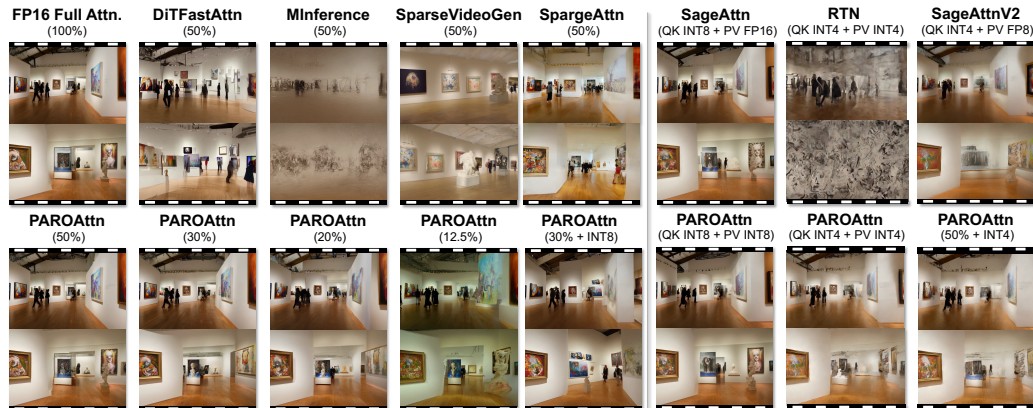

Figure 4: **Qualitative Results of CogVideoX generated videos for PAROAttention and baselines.**

dataset [38] and evaluate on a subset of prompts collected from VBench [17], covering all subtasks. For image generation, we apply PAROAttn to the Flux.1.Dev [19] model for 1024 resolution, with 30 sampling steps. The calibration prompts are the same as video generation and the first 1024 prompts from the COCO [28] dataset are used for evaluation. The block size for sparsification and quantization are chosen as 64 to align with the FlashAttention. Unlike prior work [45, 64], which avoids compressing the initial 25% of timesteps, our techniques are applied to all timesteps.

**Evaluation Metrics:** We employ two types of metrics: (1) Quality Metrics: They measure the absolute quality of videos or images. For videos, we adopt CLIPSIM [43], VQA [44], and FlowScore [32] to measure text-video alignment, quality, and temporal consistency respectively. For images, we adopt CLIPScore [13] and ImageReward [53] to measure text-image alignment, and human preference. (2) Relative Difference Metrics: They quantify the difference between FP16 generation. For both video and image generation, PSNR and cosine similarity are used to measure low-level pixel-space differences. SSIM [42] evaluates structural similarity, while FVD-FP16 [39], and FID-FP16 [14] assess feature-space differences. In practice, we find that relative difference metrics are more sensitive and better reflect the quality of compression techniques (discussed in Appendix Sec.2).

Table 2: **Performance of Flux text-to-image generation on COCO prompt set.**

| Type | Method | Efficiency | Quality | | | | |
| | | Dense Rate / Bitwidth | Image Quality Metrics | | FP Diff. Metrics | | |
| | | | CLIPScore↑ | ImageReward↑ | FID-FP16↓ | PSNR↑ | SSIM↑ | CosSim↑ |
| - | FP16 Full Attn. | 100.0% | 0.258 | 1.02 | 0.00 | ∞ | 1.000 | 1.000 |
| Sparse | DiTFastAttn (0.75) | 75.0% | 0.258 | 0.96 | 28.31 | 16.73 | 0.687 | 0.956 |
| | MInference (0.75) | 75.0% | 0.260 | 0.97 | 34.88 | 14.68 | 0.602 | 0.933 |
| | SpargeAttn (0.75) | 75.0% | 0.260 | 0.97 | 29.04 | 16.76 | 0.680 | 0.951 |
| | PAROAttn (0.75) | 75.0% | 0.259 | 1.01 | 19.47 | 20.95 | 0.812 | 0.947 |
| | DiTFastAttn (0.5) | 50.0% | 0.255 | 0.81 | 53.95 | 12.49 | 0.537 | 0.890 |
| | MInference (0.5) | 50.0% | 0.255 | 0.89 | 42.58 | 13.42 | 0.583 | 0.908 |
| | SpargeAttn (0.5) | 50.0% | 0.255 | 0.91 | 55.47 | 14.62 | 0.602 | 0.924 |
| | PAROAttn (0.5) | 50.0% | 0.259 | 1.04 | 30.39 | 16.20 | 0.683 | 0.944 |
| Quant | SageAttn | QK (INT8), PV (FP16) | 0.258 | 1.00 | 14.77 | 23.47 | 0.863 | 0.986 |
| | SageAttnV2 | QK (INT4), PV (FP8) | 0.257 | 1.00 | 20.11 | 20.95 | 0.814 | 0.979 |
| | PAROAttn (INT8) | QK (INT8), PV (INT8) | 0.258 | 1.00 | 15.34 | 23.04 | 0.856 | 0.986 |
| | PAROAttn (INT4) | QK (INT4), PV (INT4) | 0.258 | 1.00 | 19.65 | 20.16 | 0.793 | 0.975 |
| Sparse +Quant | PAROAttn (0.5+INT8) | 50% + QK, PV (INT8) | 0.259 | 1.04 | 29.56 | 16.28 | 0.680 | 0.947 |
| | PAROAttn (0.75+INT4) | 75% + QK, PV (INT4) | 0.259 | 1.01 | 22.45 | 19.26 | 0.770 | 0.971 |

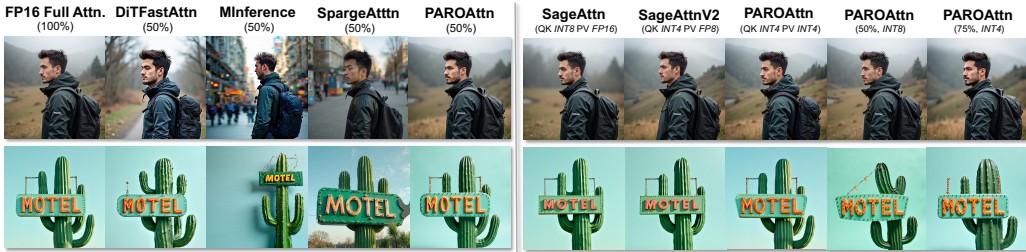

Figure 5: **Qualitative Results of Flux generated images for PAROAttention and baseline methods.**

**Baseline Methods:** For sparsification, we select baselines of different schemes, including: DiT-FastAttn [57] (dynamic window mask), SpargeAttn [62] (dynamic block-wise mask), and MInference [18], SparseVideoGen [45] (multiple static mask, dynamic selection). PAROAttn adopts static mask, and achieve superior performance under lower dense rate. For quantization, we compare with naive RTN [34], SageAttn [61] (INT8 $QK$, FP16 $PV$), and SageAttnV2 (INT4 $QK^T$, FP8 $PV$). PAROAttn quantizes $PV$ to lower bitwidth ($QK$, $PV$ INT8/INT4) with comparable performance.

**CUDA Kernel Implementation:** We implement PAROAttn based on the SageAttnV2 [59] kernel, incorporating customized designs for sparsity and quantization. Sparsification comparison are conducted on an NVIDIA A100 with CUDA 11.8, due to support limitations of baseline methods. The quantization comparison is conducted on NVIDIA RTX 4090 for FP8 and INT4 support.

## 5.2 Main Results

**Text-to-video generation:** We present the evaluation metrics in Tab. 1 and qualitative comparisons in Fig. 4, using a challenging prompt that features a complex scene with multiple objects (e.g., artworks and people). We conclude our findings as follows: (1) Baseline sparsification methods exhibit notable performance degradation for multiple metrics, even at a relatively high dense rate of 50%, resulting in visible content distortion or blurred outputs. (2) In contrast, the PAROAttn sparsification method can generate images nearly identical to the FP16 full attention baseline, even at a 20% dense rate, and achieves metric scores that surpass those of the 50% baseline. (3) The PARO token reordering is compatible with dynamic sparsity approaches. Simply combining PARO with SpargeAttn at 30% density (PSNR: 16.89) achieves performance comparable to SpargeAttn at 50% (PSNR: 16.8), yielding a speedup improvement from 1.67× to 2.22×, as shown in Fig. 6. (4) The PAROAttn quantization method maintains comparable performance while further quantizing the $PV$ to lower-bit integers (e.g., PAROAttn (INT8) vs. SageAttn, and PAROAttn (INT4) vs. SageAttnV2). (5) The PAROAttn sparsification and quantization techniques could be combined together for improved speedup. With aligned metric scores, the most aggressive plan PAROAttn (0.5+INT4) achieves nearly 10x speedup compared with baseline methods with 1.5-2x speedup.

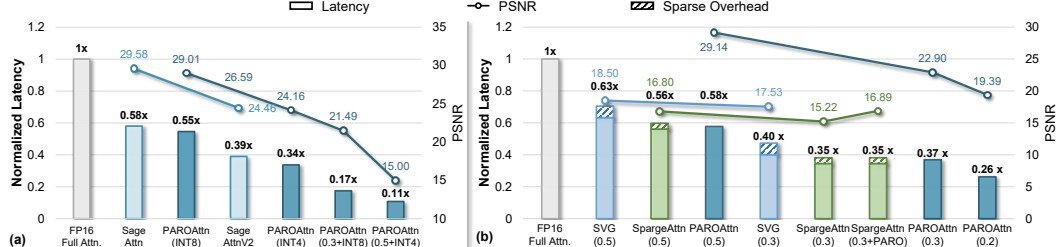

Figure 6: **Normalized latency speedup (bar plot) and PSNR (line plot)** trade-off under different (a) quantization and (b) sparse configurations.

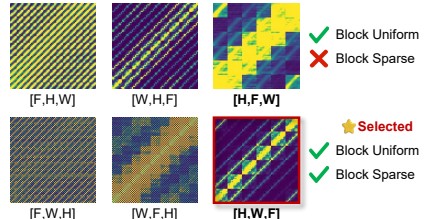

Figure 7: **Attention patterns under different permute orders.**

Table 3: **Ablation studies.** "(- Token-Reorder)" denotes PAROAttn without the token-reorder technique.

| Type | Method | FP Diff. Metrics | | | |
|---|---|---|---|---|---|
| | | PSNR↑ | SSIM↑ | LPIPS↓ | CosSim↑ |
| **Sparse** | **PAROAttn (0.5)** | 29.14 | 0.936 | 0.045 | 0.997 |
| | (- Token-Reorder) | 26.25 | 0.907 | 0.069 | 0.992 |
| | (- Timestep-Share) | 29.09 | 0.937 | 0.044 | 0.997 |
| **Quant** | **PAROAttn (PV INT8)** | 30.17 | 0.940 | 0.039 | 0.995 |
| | (- Token-Reorder) | 29.00 | 0.930 | 0.049 | 0.995 |
| | (- Block-Group) | 27.50 | 0.906 | 0.063 | 0.994 |

**Text-to-image generation:** We present the evaluation metrics in Tab. 2 and qualitative results in Fig. 5. The conclusions observed in previous sections hold consistently. Due to the shorter token lengths, sparsification becomes more challenging, and most baseline methods introduce noticeable artifacts and content distortion at a 50% density rate. In contrast, PAROAttn effectively preserves both visual quality and content, even at low density rates and when combined with quantization.

**Hardware Resource Savings:** We compare the latency speedup and performance-efficiency trade-off with baseline method's CUDA kernel implementation in Fig. 6. We conclude the key findings as follows: (1) The PAROAttn kernel achieves both superior speedup and algorithmic performance with aligned sparsity. For instance, at a 50% density rate, PAROAttn achieves a 1.73× speedup, outperforming SpargeAttn (1.67×) and SparseVideoGen (1.42×), attributed to its simplified design and reduced overhead. (2) PAROAttn's sparsification achieves speedups approaching the theoretical upper bound for computation reduction (e.g., 1.73× at 50% density, 2.71× at 30%), demonstrating its hardware-friendly nature. (3) PAROAttn introduces minimal runtime overhead (<1%), compared with SpargeAttn (6–9%) and SparseVideoGen (10–15%). (5) PAROAttn supports quantization of $PV$ to lower-bit formats (e.g., INT8/INT4), achieving similar performance to SageAttn while notably improving speedup (e.g., from 1.72× to 1.83×, and from 2.56× to 2.97×).

# 6 Analysis

We conduct extensive analyses to demonstrate the effectiveness of PAROAttn, we highlight key results here and provide additional details in the Appendix.

**Ablation Studies:** We present ablation studies of PAROAttn's techniques in Tab. 3. Removing token reorder leads to significant metric degradation for both sparsity and quantization. Similarly, eliminating timestep sharing and storing sparse masks for all timesteps does not improve performance, demonstrating that later timesteps can effectively share sparse masks. Additionally, replacing the block-wise quantization group with a row-wise approach for PV quantization results in notable degradation, highlighting the importance of the block-wise quantization group.

**Overhead Analysis:** The additional cost of PAROAttn is two-fold: The runtime overhead is minimized, as shown in Fig. 6. The offline mask generation incurs only minute-level cost, which is faster than the hyperparameter tuning required for SpargeAttn or the mask generation in SparseVideoGen.

**Effectiveness of PARO permute metric:** We visualize the attention map of the 5th head in the 1st transformer block under six different token permutation orders in Fig. 7. The permutation successfully transforms the "multi-diagonal" patterns into block-wise patterns. Notably, for the permutation $[H, F, W]$, although the values are uniformly distributed within the block, insufficient sparsity is observed. In contrast, the selected permutation $[H, W, F]$ exhibits both sparse and uniform blocks, demonstrating the effectiveness of the metrics for permutation order.

# Acknowledgement

This research was supported by National Natural Science Foundation of China (No.62203257, 62325405,62031017,62406159), Tsinghua University Initiative Scientific Research Program, Tsinghua-Efort Joint Research Center for EAI Computation and Perception, Beijing National Research Center for Information Science, Technology (BNRist), Beijing Innovation Center for Future Chips, and State Key laboratory of Space Network and Communications.

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

# A    Experimental Results for Wan 2.1 Model

We present the results of applying PAROAttn and baseline sparsification and quantization methods to the Wan-2.1[41] 14B T2V model in Tab. 4, along with qualitative results in Fig. 8. Notably, when applying SpargeAttention[63], we observed numerical instability leading to NaN outputs; hence, its results are omitted from the table. These findings are consistent with those presented for the CogVideoX model in Table 1 of the main paper. We summarize our key observations as follows:

(1) **PAROAttn consistently outperforms baseline sparsification methods across varying density.** As shown in Tab. 4, PAROAttn significantly outperforms the baseline method SparseVideoGen across different metrics and settings. Remarkably, PAROAttn with a 0.3 density still surpasses SparseVideoGen at 0.5 density.

(2) **PAROAttn preserves both visual quality and content even under more aggressive sparsity.** As illustrated in Fig. 8, PAROAttn at 0.3 density produces frames nearly identical to the dense baseline. In contrast, PAROAttn at 0.5 density introduces noticeable degradation with changes in both content and style.

(3) **PAROAttn achieves comparable performance to baseline quantization methods with higher speedups.** As demonstrated in Tab. 4 and Fig. 8, PAROAttn's quantization scheme further compresses the $PV$ computation to INT8/INT4 on top of the SageAttn baseline, maintaining performance while delivering greater speedup.

Table 4: **Performance of PAROAttention Wan 2.1 text-to-video generation on VBench prompts.** Baselines are evaluated using their official codebases. For fair comparison, we configure SparseVideoGen without skipping sparsification during the first 30% of timesteps.

| Type | Method | Efficiency | Quality | | | | | | |
|------|--------|------------|---------|---|---|---|---|---|---|
| | | Dense Rate / Bitwidth | Video Quality Metrics | | | FP Diff. Metrics | | | |
| | | | CLIPSIM↑ | VQA↑ | ΔFScore↓ | FVD-FP16↓ | PSNR↑ | SSIM↑ | CosSim↑ |
| - | FP16 Full Attn. | 100.0% | 0.215 | 93.49 | 0.000 | 0.000 | $\infty$ | 1.000 | 1.000 |
| *Sparse* | SparseVideoGen (0.5) | 50.0% | 0.199 | 91.56 | 0.468 | 0.476 | 15.32 | 0.613 | 0.900 |
| | PAROAttn (0.5) | 50.0% | 0.213 | 92.85 | 0.114 | 0.251 | 22.02 | 0.806 | 0.978 |
| | SparseVideoGen (0.3) | 30.0% | 0.196 | 90.13 | 0.612 | 0.679 | 13.17 | 0.475 | 0.839 |
| | PAROAttn (0.3) | 30.0% | 0.208 | 91.97 | 0.153 | 0.278 | 21.73 | 0.786 | 0.978 |
| *Quant* | SageAttn | QK (INT8), PV (FP16) | 0.201 | 92.24 | 0.126 | 0.209 | 20.43 | 0.720 | 0.970 |
| | SageAttnV2 | QK (INT4), PV (FP8) | 0.200 | 88.53 | 1.260 | 0.749 | 17.86 | 0.678 | 0.954 |
| | PAROAttn (INT8) | QK (INT8), PV (INT8) | 0.213 | 92.89 | 0.128 | 0.362 | 20.13 | 0.706 | 0.967 |
| | PAROAttn (INT4) | QK (INT4), PV (INT4) | 0.206 | 89.77 | 0.896 | 0.412 | 19.30 | 0.741 | 0.965 |

# B    Additional Qualitative Results and Analysis of Metrics Selection:

**Analysis of Additional Qualitative Results:** We present additional qualitative comparisons of sparsification methods, along with their corresponding metric scores, in Fig. 9. We compare PAROAttn against SpargeAttn and SparseVideoGen on the CogVideoX model under density levels of 50% and 30%. As shown in the figure, PAROAttn produces nearly identical frames at both density levels, whereas SpargeAttn and SparseVideoGen introduce noticeable blurriness and content distortion. In particular, SpargeAttn at 30% density exhibits prominent square-shaped color blocks, and the generated content becomes barely recognizable. We further analyze the corresponding changes in metric scores in the next paragraph.

**Findings and Recommendation of Metrics:** In Fig. 9, we observe that quality-related metric (VQA) and FP difference metric (PSNR) exhibit different trends as the dense rate varies. Specifically, for PAROAttn, the quality-related VQA metric remains stable even with 30% density, while the FP difference-related PSNR metric gradually decays. It reveals that the FP difference-related metrics are more challenging to maintain because they focus on low-level differences and can detect very minor detail changes that quality-related metrics might miss. As a result, they are suitable for scenarios where only minor differences are expected. However, they may not be reliable when comparing samples with significant differences, in which case quality-related metrics are more appropriate.

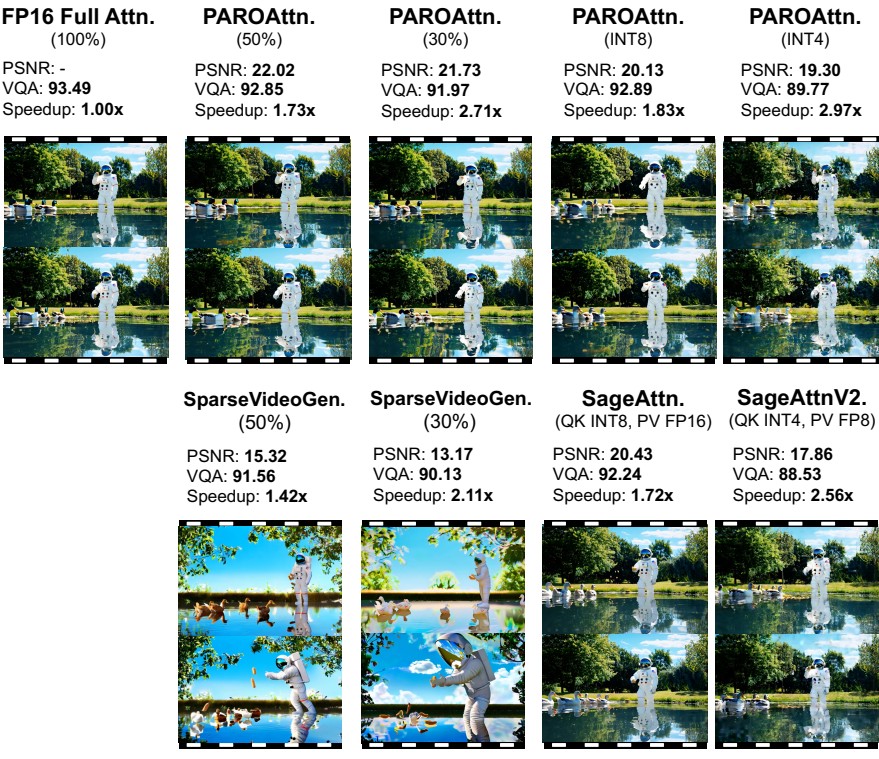

Figure 8: **Qualitative results of Wan 2.1 model video generation.**

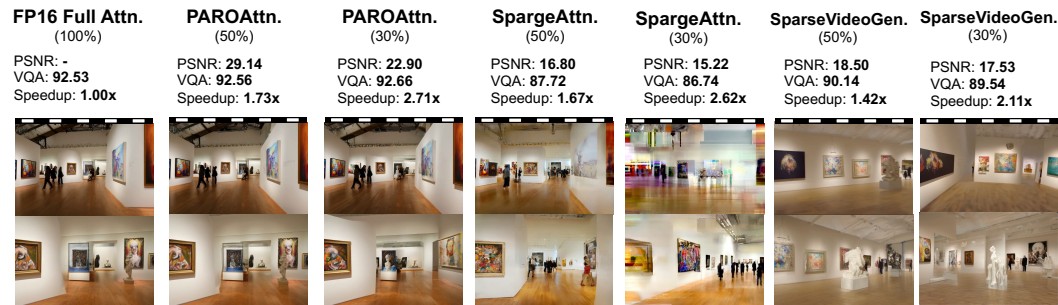

Figure 9: **Additional qualitative results of sparsification for CogVideoX.**

**Ablation study on skipping scheme in SparseVideoGen.** In the original SparseVideoGen paper and its official code release, sparsification is deliberately omitted for the first two Transformer blocks and the initial 30% of timesteps. In the main paper, for fair comparison, we donot adopt such "skipping" scheme for SparseVideoGen. We conduct an ablation study on the effect of this "skipping" scheme, with results shown in Sec. B and Fig. 11. As observed, removing the skipping strategy leads to a notable degradation in generation quality (PSNR drops from 25.37 to 18.50), significant content distortion, and visibly blurred outputs. In contrast, PAROAttn maintains high generation quality even without skipping early timesteps or transformer blocks.

## C  Additional Results for CUDA Kernel Efficiency Improvement

We provide detailed results comparing different CUDA kernel implementations in Sec. C and Sec. C. The reported speedups are measured based on attention computation alone (excluding the QKVO projections), using a token length of 17,750—corresponding to 6-second 720P video generation with CogVideoX. Experiments for sparsification baselines are conducted on an NVIDIA A100 GPU

| FP16 Full Attn. (100%) | PAROAttn. (50%) | SparseVideoGen. (50%, w. Skip) | SparseVideoGen. (50%, w.o. Skip) |
|---|---|---|---|
| PSNR: -
VQA: **92.53**
Speedup: **1.00x** | PSNR: **29.14**
VQA: **92.56**
Speedup: **1.73x** | PSNR: **25.37**
VQA: **91.89**
Speedup: **1.42x** | PSNR: **18.50**
VQA: **90.14**
Speedup: **1.42x** |

| Method | PSNR↑ | SSIM↑ | CosSim ↑ |
|---|---|---|---|
| SparseVideoGen (0.5, w.o. skip) | 18.50 | 0.755 | 0.960 |
| SparseVideoGen (0.5, w. skip) | 25.37 | 0.871 | 0.984 |
| PAROAttention (0.5) | 29.14 | 0.936 | 0.997 |

Figure 10: **Ablation of skipping timestep and transformer blocks for SparseVideoGen.**

Figure 11: **Qualitative examples from the ablation study on skipping timesteps and Transformer blocks in SparseVideoGen.**

(consistent with the supported hardware in the baseline code release), while quantization results are evaluated on an RTX 4090 GPU to leverage support for INT4 and FP8 quantization.

**Comparison of Latency Speedups**: We detailedly present the experimental results for PAROAttn's CUDA kernel implementation with baseline sparsification and quantization methods in Sec. C. We discuss the findings in Sec. 5.2 "Hardware Resource Savings" of the main paper in detail as follows:

- **Baseline Sparsification methods exhibit notable performance degradation.** Both the SparseVideoGen and SpargeAttn achieve PSNR lower than 20, even with a relative high density of 50%, worse than the PAROAttn with both sparsification and quantization applied (0.3+INT8 with PSNR 21.49, 0.5+INT4 with PSNR 24.34).

- **PAROAttn's sparsification method can generate nearly identical frames, even with lower dense rate.** As seen in Sec. C and Fig. 9, the PAROAttn generation result highly resembles the full-precision baseline, while achieveing substantial speedups.

- **The PARO token reordering is compatible with dynamic sparsification approaches.** In Sec. C, the "SpargeAttention (0.3 + PARO)" means adopting the PARO token reordering with SpargeAttn, it notably improves the performance to exceeding the "SpargeAttn (0.5)", and improve the speedup from 1.67x to 2.11x.

- **PAROAttn introduces minimal overhead.** The overhead of sparsification is presented in Sec. C, since the PAROAttn adopts static sparse scheme, it avoids the online static mask generation overhead. The remaining overhead is the online permutation, and loading of the sparse mask, which are also diminished with the kernel fusion and prefetch techniques, discussed further in the "overhead of permutation/prefetch" paragraph below.

- **PAROAttn supports quantization of PV to lower-bit formats** Comparing the PAROAttn (INT8/4) with SageAttn (V1/V2), PAROAttn could further quantizes the $PV$ computation from FP16/FP8 to INT8/INT4 with similar performance, and notable better speedup (1.72x to 1.83x, and 2.56x to 2.97x).

**Overhead of Permutation:** We present an overhead analysis of integrating permutation within the Rotary Position Embedding (RoPE) operator. As shown, this integration introduces only negligible overhead (0.03%), demonstrating that permutation can be fused with prior operators without performance impact.

**Overhead of Prefetch:** As discussed in Section 4.2 of the main paper, static sparse attention introduces additional memory overhead due to the need to store the sparse mask in GPU memory. Since we adopt timestep-wise and transformer-block-wise sparse masks, the memory cost of storing the binary sparse mask is approximately 1GB. To mitigate this cost, we introduce a prefetch scheme that only loads the sparse mask for the current transformer block and timestep, reducing memory usage to the KB level. Additionally, we employ a double-buffering pipeline technique that allows for simultaneous sparse mask loading and attention computation, minimizing the time spent on sparse mask loading. Overall, the prefetching incurs only 0.33% of the total latency.

| Method | PSNR↑ | Speedup↑ | Overhead ↓ |
|---|---|---|---|
| FlashAttention | - | 1.00x | - |
| SpargeAttention (0.5) | 16.80 | 1.67x | 6% |
| SparseVideoGen (0.5) | 18.50 | 1.42x | 10% |
| PAROAttention (0.5) | 29.14 | 1.73x | 0% |
| SpargeAttention (0.3) | 15.22 | 2.62x | 9% |
| SpargeAttention (0.3 + PARO) | 16.89 | 2.62x | 9% |
| SparseVideoGen (0.3) | 17.53 | 2.11x | 15% |
| PAROAttention (0.3) | 22.90 | 2.71x | 0% |

Table 5: **Comparison of latency speedup for sparsification methods on NVIDIA A100.**

| Method | PSNR ↑ | Speedup ↑ |
|---|---|---|
| FlashAttention | - | 1.00x |
| SageAttnV1 | 29.58 | 1.72x |
| PAROAttn (INT8) | 29.01 | 1.83x |
| SageAttnV2 | 24.46 | 2.56x |
| PAROAttn (INT4) | 24.16 | 2.97x |
| PAROAttn (0.3 + INT8) | 21.49 | 5.72x |
| PAROAttn (0.5 + INT4) | 24.34 | 9.28x |

Table 6: **Comparison of latency speedup for quantization methods on NVIDIA RTX4090.**

Table 7: **Overhead of permutation.** The latency comparison of whether adopting permutation to rope operator. The "w." and "w.o." stands for with and without

| | w.o. permute | w. permute | overhead |
|---|---|---|---|
| Latency (ms) | 1.2488 | 1.2492 | 0.03% |

## D  Implementation Details and Analysis of Baseline Sparsification Methods

**Implementation details:** We select MInference [18], DiTFastAttn [57], SparseVideoGen [46], and SpargeAttention [63] for video generation. We visualize the sparse mask generated by baseline sparse methods in Fig. 12.

- **MInference:** We adapt the sparse attention scheme designed for language models to visual attention masks, making the following modifications: First, since we skip sparsification for the larger text tokens, the "attention sink" phenomenon—where the first few tokens are significantly larger than the rest—is not observed. As a result, the "Δ-shaped" pattern degrades to a single diagonal pattern, which can be viewed as a special case of the "vertical-slash" pattern. We select between the remaining "vertical-slash" and "block-wise" patterns based on cosine similarity, following the original implementation. Consistent with the original paper, the "vertical-slash" pattern is determined by selecting the top $K\%$ of vertical and diagonal lines with the largest summed values, where $K$ can be tuned to adjust the sparsity rate. For the "block-wise" pattern, we use $8\times8$ blocks and determine whether to retain each block based on its summed value.

- **DiTFastAttn:** Consistent with the original paper, we determine the window length by selecting the smallest window where the sum of values within the window reaches $K\%$ of the total attention values.

- **SparseVideoGen:** We use the official code implementation, the num-sampled-rows are chosen as the default value 32 and 64 for CogVideo and Wan. Specifically, for fair comparison, we donot adopt skipping sparsification for the first timesteps, and set first-times-fp as 0. We also present the ablation of such skipping scheme in Sec. C.

- **SpargeAttn:** We use the official code implementation to test the performance of the CogVideoX model. When adapting SpargeAttn to Wan, a numerical stability issue arises, causing the attention computation to produce NaN values; therefore, we omit these results. The hyperparameters for SpargeAttn are tuned using the script provided in the official code.

Table 8: **Overhead of prefetching.** The latency comparison of whether adopting prefetch for attention.

| | w.o prefetch | w. prefetch | overhead |
|---|---|---|---|
| Latency (ms) | 1296.5 | 1300.8 | 0.33% |

For a density of 50%, we set $l1 = 0.09$ and $pv_{l1} = 0.095$. For a density of 30%, we choose $l1 = 0.13$ and $pv_{l1} = 0.135$.

**Visualization of attention masks:** We present a comparison of sparse masks for PAROAttention and baseline sparse attention methods. The first column shows the post-softmax attention patterns. As can be seen, for the SparseVideoGen method, while it successfully identifies the "diagonal in block" temporal attention pattern, the diagonal selection within the block remains inaccurate even at a relatively high dense rate. In contrast, PAROAttn effectively preserves attention values while exploiting sparsity. For DiTFastAttn, the window-based attention struggles with the multiple diagonal pattern and fails to capture diagonals located far from the center. Similarly, MInference's diagonal pattern is unable to accurately preserve the naturally block-wise attention pattern.

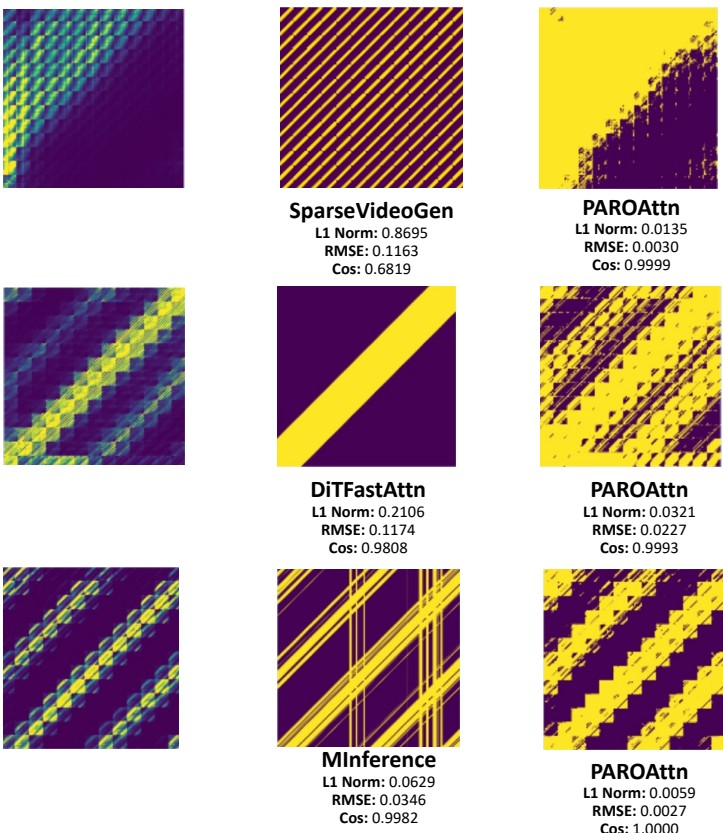

Figure 12: **Comparison of attention masks for PAROAttention and baseline sparse attention methods.** We present the relative difference metrics (L1 Norm, RMSE, CosSim) to measure the difference between the original and masked attention map.

# E   The Effectiveness of PAROAttention Quantization Technique

**Incoherence Analysis:** To demonstrate the effectiveness of PAROAttention's quantization technique, we present the data distribution within the quantization group (a 64×64 block) in Fig. 13. As shown, similar values are successfully aggregated into localized blocks, and the outliers present in the original data distribution are significantly reduced. This reduces the incoherence range from 200-1200 to 12-20, and thus significantly reducing the quantization error.

**Comparison with FP8 quantization:** In SageAttnV2 (Table 6), the authors analyze the cosine similarity between the quantized $P$ matrix and its original FP counterpart, concluding that INT8 quantization introduces too much error and thus opting for FP8. However, after applying pattern-aware token reordering, the incoherence within the attention map data groups is significantly reduced,

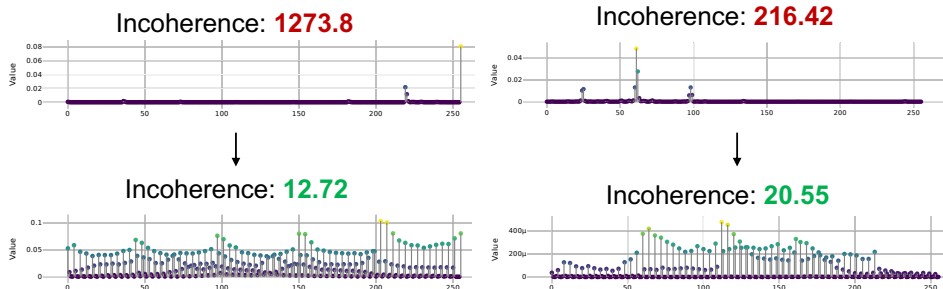

Figure 13: **Incoherence for data within the quantization group before and after permutation.**

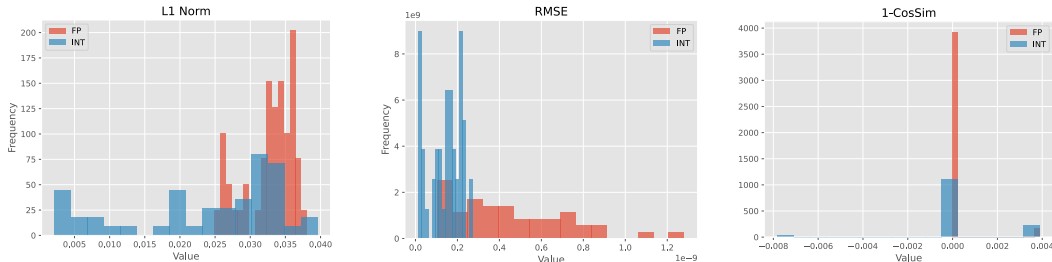

Figure 14: **Quantization error with respect to FP for quantization of the attention map $P$.** The red "FP" stands for the FP8 quantization error.

leading to a notable decrease in quantization error. As shown in Fig. 14, the INT8 quantized attention map achieves significantly higher FP difference metric scores compared to its FP8 counterpart. This is because INT8 provides more mantissa bits to accurately represent subtle value differences.

**Reasons for exploring interger quantization:** Despite FP8 quantization achieves good performance and valid acceleration with easy deployment. We summarize the reasons for exploring integer quantization as follows:

- **Lower quantization error:** Low-bit floating-point formats consist of both exponent bits and mantissa bits. For example, the E5M2 FP8 format has 5 exponent bits and 2 mantissa bits. The reduced number of mantissa bits limits its ability to represent small value differences, potentially leading to performance degradation. In contrast, with the same bitwidth, integer formats provide more mantissa bits (e.g., equivalently 7 mantissa bits for INT8), enabling them to preserve subtle data differences and achieve lower quantization error. By applying proper preprocessing to remove outliers within data groups, the need for additional exponent bits to handle large dynamic variations is reduced. This advantage becomes even more pronounced at lower bitwidths, such as 4-bit, where FP4 formats have only 1-2 mantissa bits. As presented in Fig. 5 in the main paper, the **all INT4** PAROAttention quantized Flux model could still generate images with high quality. To conclude, integer quantization's representation power is essential for lower bitwidth quantization.

- **Support non-GPU hardware platforms.** Despite Nvidia TensorCore [1] demonstrate simialr computing power for FP8 and INT8 matrix multiplication. However, for domain-specific accelerator design [2], adopting INT8 matrix multiplication could be more resource-efficient than FP8. Therefore, integer quantization is valuable for AI accelerator hardware design beyond GPU.

## F Generalization of Sparse Attention Mask

We present a visualization of the post-softmax attention patterns across different timesteps, prompts, and classifier-free guidance (CFG) settings in Fig. 15. The relative metric scores (L1 Norm, RMSE, cosine similarity) are calculated based on the attention pattern at timestep 5, as indicated by the red text. As shown, the type of attention pattern remains consistent across timesteps, prompts, and

CFG. However, the detailed attention pattern may vary over timesteps, gradually stabilizing in later timesteps. To address this, we design timestep-wise sparse masks and share the sparse mask for later timesteps.

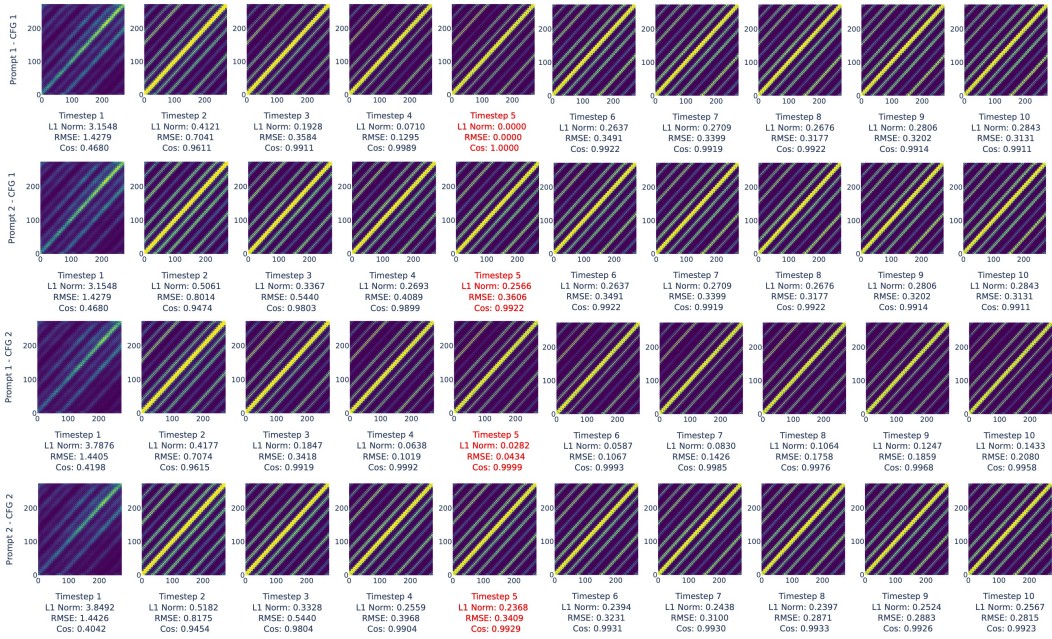

Figure 15: **Visualization of post softmax attention pattern for different timesteps, prompts, and classifier-free-guidance (CFG).** The metric scores are calculated relative to the attention pattern with red text.

# G   Additional Visualization of Permutation for Flux

We present the attention pattern for flux under different permutations. The permutation also effectively produces concentrated and regular block-wise pattern.

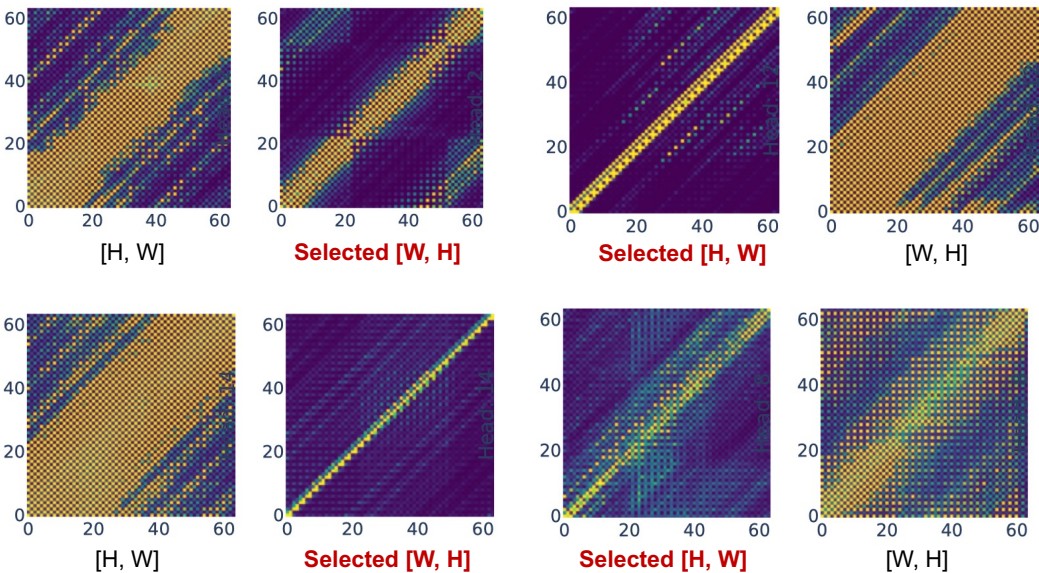

Figure 16: **Visualization of the attention pattern for flux under different permutation.**

# H Discussion of application of PAROAttn

As discussed in the "Discussion of Adaptability" section of the main paper, we introduce pattern-aware token reordering (permutation) as a universal and efficient preprocessing step for attention patterns. Its effectiveness stems from the unique properties of vision transformers, where 3D (or 2D) spatial information is flattened into a 1D token sequence, disrupting local adjacency. By concentrating attention patterns into more regular and block-wise structures, it benefits a wide range of application scenarios.

**For Compression Techniques:** The advantages of this approach extend beyond the specific design of PAROAttention and are applicable to various compression techniques, such as timestep-wise sharing [30], efficient reasoning [68, 10, 12], efficient architecture design [25, 52, 47]. For dynamic sparse attention methods, such as SpargeAttn [63], the relative importance of blocks becomes more apparent, simplifying the task of generating sparse masks from QK embeddings. Additionally, the concentrated patterns can improve caching strategies for feature reuse.

**For Model Training:** PAROAttention's permutation design also sheds light on the meaning of attention patterns, which could inspire future improvements in model training. For instance, it could encourage different attention heads to focus on aggregating information along different dimensions, leading to more specialized and efficient learning.

**Beyond Visual Generative Models:** The effectiveness of permutation arises from the unique properties of vision transformers, but its applicability is not limited to visual generative models. It could potentially be extended to multi-modal large language models and large vision models for perception tasks, offering similar benefits in these domains.

# I Limitations and Broader Impacts

The methodology can be further improved from several perspectives. Permutation represents a constrained subset of possible token reordering, and we adopt simple block sum as sparse metric, exploring more advanced reordering techniques or sparse metric could further enhance performance. PAROAttn introduces a novel direction by leveraging token reordering to reorganize attention patterns. The idea is not limited to post-training compression, and could be extended to broader applications, such as enabling native sparse attention or training acceleration.

