# OpenReview forum: "PAROAttention: Pattern-Aware ReOrdering for Efficient Sparse and Quantized Attention in Visual Generation Models"
_NeurIPS.cc/2025/Conference — NeurIPS 2025 poster_

### Official Review · Reviewer_A1jm · 2025-07-01

**Clarity:** 3
**Significance:** 3
**Originality:** 3
**Rating:** 5
**Confidence:** 3

**Summary:**

This manuscript propose a novel pattern-aware token reorder (PARO) method for efficient sparse and quantized attention. The authors provide a key insight of the challenges of the attention sparsification and quantifization, and proposed PAROAttention to reorganize and unify the attention. Experimental results the proposed method achieves lower density and bitwidth, and achieving a higher end-to-end latency speedup.

**Questions:**

Overall, this is a solid method. The following are my major concerns:
1. The guarantee of the selected permutation. Do the 6 permutations definitely contain a case that minimizes the diversity of the attention variants under a certain value? If yes, are there empirical or theoretical justifications?
2. The generalization of the proposed method. Is the proposed method applicable to other attention-based methods on different tasks? If not, please justify the reasons.
3. The ability boundary of the proposed method is expected to be justified.

I'd like to raise my rating if my major concerns are addressed.

**Ethical Concerns:**

["NO or VERY MINOR ethics concerns only"]

**Final Justification:**

The authors have addressed my concerns about the method in the worst-case scenarios

**Limitations:**

The generalization and ability boundary of the proposed method are expected to be justified.

**Quality:**

3

**Strengths And Weaknesses:**

Strength:

1.	The observation and motivation of the proposed method are interesting.
2.	The proposed method PARPAttention, is hardware efficient.
3.	Experiment results demonstrate good performance.

Weakness:

1.	A more detailed explanation of the figures is expected. For example, in Figure 2 (b), why is the left attention map hard to sparsify and quantize?
2.	For the pattern-aware reordering, is there a guarantee, such as sparsity, on the best permutation choice from the 6 possible cases? What is the worst case of the chosen best permutation?

---

> ### Author Rebuttal · Authors · 2025-07-31
>
> We sincerely thank reviewer A1jm for the time and efforts, constructive feedback, and insightful suggestions, which are invaluable for improving our paper. We are encouraged by the reviewers’ recognition of the motivation, hardware efficient and solid performance improvement. We address each of the raised weaknesses and questions as follows:
>
> > **Weakness 1:** *More detailed explanation of Fig.2, why left one is hard to sparsify and quantize.*
>
> **Response to W1:** We thank the reviewer for pointing out the clarity issue in the explanation of Fig. 2. The explanation is provided in Section 3 of the main paper, and we will discuss it in detail here and reorganize the description in a future version.
>
> - **Sparsity:** As discussed in Line 103, Section 3 of the main paper, structured sparsity is essential for achieving practical speedup. The lefter attention patterns in Fig.2  are **diverse in shape**, which requires a complex sparse mask design to match them accordingly. Additionally, the **large attention values are scattered and dispersed**, resulting in a lack of complete sparse local regions, which prevents the formation of hardware-friendly structured sparsity.
> - **Quantization:** As discussed in Line 117, Section 3 of the main paper, prior research [1] summarizes that the major source of quantization error arises from large variations within data groups. Since **the large attention values are scattered and dispersed**, significant variation exists within local data groups, which, in turn, increases quantization error.
>
> [1] Zhao et. al., ViDiT-Q: Efficient and Accurate Quantization of Diffusion Transformers for Image and Video Generation, arxiv 2406.02540
>
> ---
>
> > **Weakness 2:** *Is there a guarantee of sparsity 6 permutation choices? What is the worst case of chosen best permute?*
>
> > **Q1:**  *Empirical or theoretical justifications The guarantee of the selected permutation*
>
> **Response to W2 and Q1:** We agree with the reviewer that the selection and effect of reorder strategy is a key issue. We first provide direct answer to the raised 2 questions, then provide analysis of the reorder strategy design.
> - ***"Is there a guarantee of sparsity 6 permutation choices?":*** Due to the naturally diverse patterns, there is no theoretical guarantee for the sparsity rate. However, the PARO reordering plan ensures the selection of the **"relatively better"** plan through simple enumeration of the carefully crafted search space. Since the search space extensively captures the local aggregation characteristics, which are inherent to visual feature extraction, it is **highly likely to contain a good sparse case**. This approach has been **empirically proven effective** in producing structured sparsity, as demonstrated by the visualization examples in Appendix Figures 5 and 8, and the experimental results in the "Discussion of reorder strategy" section below.
> - ***"What is the worst case of chosen best permutation?":*** For some cases, the simple permutation has room for improvement, such as presented in lower-right Fig. 9.
> - **Discussion of the reorder strategy:**
>   - The current reordering design is relatively simple and serves as a **proof of concept to validate its effectiveness**, leaving room for improvements to further enhance its potential.
>   - The design of PAROAttn **is compatible with more advanced optimization strategies for token reorder**, we adopt a static approach that determines both the sparsity and reordering strategies offline. This design enables more flexible and sophisticated optimization, as it avoids incurring additional online overhead.
>   - To further demonstrate this potential, we supplement our experimental results with an extension that integrates the existing PAROAttn framework with **balanced K-Means clustering**.
>
> |   Method (CogVideo)      | PSNR (videos) | SSIM (videos) |
> |:-------:|:-------------:|:-----------:|
> | PARO      |     19.61     |     0.753     |
> | PARO+Cluster     |     20.37     |     0.790     |
>
> As can be seen, combining balanced clustering with the current PAROAttn yields only a slight performance improvement. These results empirically validate both **the effectiveness of the current PARO strategy** and its **compatibility with more advanced optimization techniques**.
>
> ---
>
> > **Q2:** *Is PAROAttn applicable to other attention-based methods for different tasks?*
>
> **Response to Q2:** Yes, the redundancy of global attention in visual feature extraction tasks with a local aggregation nature is a common problem. The PAROAttn token reordering technique is applicable to **a variety of tasks beyond visual generation, such as  multimodal understanding (vision language models).**
>
> We apply PAROAttn to **LLaDa-V [2]**, a diffusion-based vision-language model, to demonstrate its applicability beyond visual generation. We select the MathVerse dataset because it involves high-resolution images, and the image processing accounts for the majority of the computational cost.
>
> | Methods | Mathverse-minivision Score  |
> |:----:|:----:|
> |  Origin    |   28.5         |
> |  Naive (0.75)    |   20.5         |
> |  PAROAttn (0.75)    |   27.8         |
>
> [2] You et. al., "LLaDA-V: Large Language Diffusion Models with Visual Instruction Tuning", arxiv 2505.16933
>
> Beyond diffusion-based methods, Similar "mutliple-diagnoal" pattern also exists for the prefilling of VLMs, as illustrated in the figures in the papers below (We apologize for the indirectness since NeurIPS rebuttal does not support providing figure or links).
> - Right side of Figure 6 "BlindSight: Harnessing Sparsity for Efficient VLMs" (arxiv 2507.09071)
> - Figure 3 of "MMInference: Accelerating Pre-filling for Long-Context VLMs via Modality-Aware Permutation Sparse Attention" (arxiv 2504.16083)
>
> ---
>
> > **Q3:** *Justification of the ability boundaries of PAROAttn*
>
> **Response to Q3:** We discuss the ability boundary of PAROAttn from the following aspects:
> - **Performance Upper Bound:** For standalone sparsity and quantization, as shown in Fig. 4 of the main paper, a 12.5% dense rate (8x sparse) causes a noticeable color shift, and INT4 leads to slight jittering across frames, indicating approaching performance boundary. When combining sparsity and quantization, "30% dense rate + INT8" and "50% + INT4" represent boundary cases that extensively preserve generation quality, achieving 5.8x and 9.1x end-to-end attention speedup, respectively. More aggressive compression plans are expected to cause notable degradation, except in extreme application scenarios, such as very long sequence generation, where there is inherently more redundancy.
> - **Application Field:** As discussed in Q2, PAROAttn addresses the generic problem of exploiting spatial redundancy for visual feature extraction. It can be widely applied not only to diffusion-based visual generation but also to the pre-filling stage of autoregressive models and multimodal understanding tasks. It is suitable for **any application scenario that requires compressing 'inter-image' attention**, but may need adaptation for attention mechanisms that describe 'text-image interaction' or the decoding stage of autoregressive models, which may exhibit distinct attention pattern characteristics.
> ---
>
> > **Limitations:** *The generalization and ability boundary of the proposed method*
>
> **Response to Limitations:** We detailedly discuss the generalization of the method in Q2, and discuss the ability boundary in Q3.
>
> ---
>
> We sincerely thank the reviewer for the time and efforts. We hope our responses have addressed your concerns. Please don’t hesitate to let us know if you have any additional questions or feedback on our response. We will do our best to address any concerns you may have :)

---

> > ### Comment · Reviewer_A1jm · 2025-08-03
> > **Have addressed my concerns**
> >
> > Thanks for the detailed responses, and I will update my rating

---

> > > ### Author Response · Authors · 2025-08-03
> > > **Gratitude for Feedback and Acknowledgment**
> > >
> > > Thank you for the update. We truly appreciate your acknowledgment of our contributions and response. We are sincerely grateful for your constructive feedback and the time you invested in reviewing our paper.

---

### Official Review · Reviewer_CDie · 2025-07-01

**Clarity:** 3
**Significance:** 3
**Originality:** 3
**Rating:** 4
**Confidence:** 4

**Summary:**

This paper proposes PAROAttention, a novel and efficient sparse and quantized attention mechanism for visual generation models based on the token reordering technique. By unifying diverse attention patterns into block-wise structures, PAROAttention makes attention sparsification and quantization more hardware-friendly and computationally efficient. Specifically, this work first highlights that the primary challenge in compressing attention in visual generation models arises from the diverse and scattered distribution of visual attention. Then, it further finds that these unique patterns result from the local aggregation of visual information across different dimensions, which can be transformed into localized block-wise structures by gathering the corresponding tokens through token reordering. Moreover, this reordering operation groups similar attention values together, simplifying quantization by eliminating the influence of outliers. Because of the block-wise design, the sparsification granularity naturally aligns with FlashAttention implementations. Consequently, PAROAttention achieves high-quality video generation under aggressive compression settings, including 20% sparsity and INT4 quantization, and delivers up to 3.8× end-to-end speedup, outperforming existing attention compression methods for visual generation models.

**Questions:**

(1) Table 3 illustrates that removing timestep-share cannot achieve better performance, which is kind of counterintuitive: with precise mask for each step, the performance is supposed to be better. Could the authors please try to explain this interesting results?

(2) In the figure3, it seems that finally we need to do an inverse-reorder operation to recover the original sequence. However, there is no further elaboration in the paper for this operation. Is there anything important to note for this operation such as the overhead of this operation? Will the performance degrade a lot if we do not do this inverse-reorder?

(3) PAROAttention harnesses the offline determined permute order. Are these determined permute patterns model-specific? Moreover, it claims that this offline mask generation is minute-level cost, is this because for different model and different input visual information, we need to do this offline mask generation from scratch?

(4) How does this reorder-then-compress could potential benefits the large vision language models? could you please share some detailed thinkings about this? I hope to see that this idea can have broader impacts beyond visual generation domain.

**Ethical Concerns:**

["NO or VERY MINOR ethics concerns only"]

**Final Justification:**

After rebuttal and discussions with authors, my final results can be summarized as follows:

**Weaknesses**

For the concerns about the permutation strategy design space and generalization ability, the authors reply make sense and make me convinced about the design choice;

For the concerns about the parameter Sensitivity for reorder strategy and sparse block size, the authors' motivation is clear and the results make sense.

**Questions**

All the questions are answered convincingly by the authors, and the extra experiments on LLada further proves the effectiveness of PAROAttention design.

**Summary**

Overall, I think this work is robust and impact, and it reaches to the requirement of NeurIPS. Therefore, I give it borderline acceptance.

**Limitations:**

The major limitations are mainly about the static, offline permutation patterns determination and the sparsity metrics, which are elaborated in the weaknesses and questions above.

**Paper Formatting Concerns:**

No paper format concerns.

**Quality:**

3

**Strengths And Weaknesses:**

**Strengths:**

**(1) Method:** The core idea of reordering tokens is novel and interesting for compressing attention in visual generation models. It opens up a new perspective on attention compression by avoiding the need to design complex sparsification and quantization methods to adapt to the diverse attention patterns typically observed in visual data.

**(2) Experiments:** The experimental setup in the paper is clear and make sense. The authors provide a comprehensive evaluation across different settings and baselines on both video and image generation tasks. Compared to existing methods, including SageAttn and SparseAttn, PAROAttention achieves high generation quality under extreme compression scenarios, such as 20% density and INT4 quantization for both QK and PV, while delivering up to 3.8× end-to-end speedup. The ablation studies are convincing as well.

**(3) Practicality:** The work also includes the design of specialized CUDA kernels to support PAROAttention, making the approach practical and accessible for evaluation by the community.

**(4) Writing:** Overall, the structure and style of the paper are clear and easy to follow, despite some minor typos (e.g., line 155: “to to”) and some missing details. The figures effectively illustrate the methods and results.

**Weaknesses:**

**(1) Permutation Generality:** The paper claims that the six possible permutations are sufficient to produce unified, block-wise patterns. However, there is no thorough analysis to support this claim. This raises doubts about whether these six static permutations are adequate for handling the diverse and changing content across different samples and domains and visual generation models. This may be one reason the model’s performance could be constrained after compression. Additional discussion and empirical analysis of how permutation patterns influence effectiveness would strengthen the work.

**(2) Parameter Sensitivity:** The parameter sensitivity is not fully elaborated in the paper. For example, the metrics for permutation selection involve several hyperparameters, but it is unclear how sensitive the chosen patterns are to these settings, which is important to understand. Moreover, the block size is set to 64 to align with FlashAttention, but whether 64 is the optimal value remains unexplored. If 64 is too large to effectively capture local information, reducing the block size might further improve accuracy. A more detailed analysis of how block size and other parameters affect performance would be valuable.

---

> ### Author Rebuttal · Authors · 2025-07-31
>
> We sincerely thank reviewer CDie for the time and efforts, constructive feedback, and insightful suggestions, which are invaluable for improving our paper. We are encouraged by the reviewers’ recognition of the novelty of methodology design, comprehensive evaluation, practicality, and clear writing. We address each of the raised weaknesses and questions as follows:
>
> > **Weakness 1:** *Permutation strategy design space and generalization ability.*
>
> **Response to W1:** We agree with the reviewer that selecting the appropriate reordering (permutation) strategy is fundamental to PAROAttn, and that providing more analysis on this topic would be beneficial.
>
> - **Design space of the reorder strategy.** *"Whether the six possible permutations are sufficient to produce unified, block-wise patterns."*
>   - The current reordering design is relatively simple and serves as a **proof of concept to validate its effectiveness**, while leaving room for future improvements to further enhance its potential.
>   - **The current PARO plan is empirically verified effective despite simplicty:** The current PARO design is guided by the strong prior of the local aggregation nature inherent in visual feature extraction. Despite the simplicity of the design and the limited search space, PARO demonstrates strong pattern reorganization capabilities, effectively generating unified block-wise patterns in most cases, as illustrated in Appendix Figures 5 and 8.
>   - **The design of PAROAttn is compatible with more advanced optimization strategies for token reordering.** PAROAttn adopts a static approach that determines both the sparsity and reordering strategies offline. This design enables more flexible and sophisticated optimization techniques, as it avoids incurring additional online overhead. To further demonstrate this potential, we supplement our experimental results with an extension that integrates the existing PAROAttn framework with balanced K-Means clustering.
>
> |   Method (CogVideo)      | PSNR (videos) | SSIM (videos) |
> |:-------:|:-------------:|:-----------:|
> | PARO      |     19.61     |     0.753     |
> | PARO+Cluster     |     20.37     |     0.790     |
>
> As can be seen, combining balanced clustering with the current PAROAttn yields only a slight performance improvement.  These results validate both **the effectiveness of the current PARO strategy** and its **compatibility with more advanced optimization techniques.**
>
> - **Generalization of the Reorder Strategy:** *"Are these six static permutations sufficient for handling the diverse and changing content across different samples, domains, and models?"* We carefully investigate the generalization of the reorder strategy and sparse mask across multiple dimensions.
>   - **(1) Models:** Despite the diverse and changing patterns across models, the six permutations are designed based on a strong prior of the local aggregation nature of visual feature extraction. We present visualizations of the attention patterns for the CogVideo and Flux models in Appendix Fig. 8 and 9, which reveal a clear block-wise pattern, though leaving room for further improvement.
>   - **(2) Timesteps:** We observe that attention patterns vary across timesteps and adopt timestep-wise sparse masks. However, attention patterns of different timestep share similar trend in local aggregation dimension (as seen in Appendix Fig.8). Therefore, we choose the same permute plan across timesteps, but verify the effectiveness of current reorder plan.
>   - **(3) Conditions (Prompt and Random Noises):** The attention patterns exhibit high similarity (CosSim > 0.99) across different conditions, demonstrating that the reorder strategy and sparse mask generalize well. We further supplement our experiments with a new prompt set (VBench 2) and random noise, and the performance remains strong.
>
> | Method (VBench2)        | SSIM (video)             | PSNR (video)             |
> |:---:|:----:|:----:|
> | PAROAttn (0.5) | 0.919 (±0.0161)    | 28.442 (±1.054)     |
>
> ---
>
> > **Weakness 2:**  *Parameter Sensitivity for reorder strategy and sparse block size.*
>
> **Response to W2:** We agree with the reviewer and provide an analysis and justification for our choice of hyperparameters.
> - **Hyperparameters for permutation selection:** Since the final task is a simple 1-out-of-6 classification, **the selection of hyperparameters is relatively robust**. We conducted additional experiments and found that varying the `sparse_percentage` from 75% to 95% and the `sparse_threshold` from 1e-4 to 5e-3 does not affect the permutation selection. The `alpha` is used to control the trade-off between sparsity and quantization, the final permutation plan only changes when alpha is close to 0.5.
> - **Block size selection:** The block size of 64 is selected **based on actual profiling.** Intuitively, speedups are positively correlated with block size. However, in practice, when the block size is relatively large and the sequence length is not long enough (e.g., in our case with a sequence length of 17K), larger block sizes, such as 128, perform worse than 64. This is due to the "spilling" phenomenon, where the cache and register usage for each block may exceed the hardware limits, causing the CUDA compiler to move some data to lower-speed memory, which sacrifices efficiency.
>
> | | Block 32 | Block 64 | Block 128 |
> |:-------:|:----:|:-------:|:-----:|
> | Seq 17K     | 2.91           | **3.55**           | 3.54           |
> | Seq 32K     | 2.97           | 3.63           | **3.79**          |
>
> ---
>
> > **Q1:** *Why removing timestep sharing achieve better performance.*
>
> **Response to Q1:** Removing timestep sharing is expected to yield improved results, at the cost of increased resource consumption. The phenomenon where some metrics (specifically PSNR) appear lower, while SSIM and LPIPS improve, is **due to evaluation noise**, as the performance differences are minimal and difficult to discern by human observers. As seen in Appendix Fig. 8, the attention patterns for later timesteps are very similar, thus could be shared.
>
> ---
>
> > **Q2:** *Essentialness of the inverse order and the overhead*
>
> **Response to Q2:** The inverse reordering process is essential to maintain mathematical equivalence with the reordering process. We omit the implementation details of the inverse reordering process as **it is analogous to the reordering process**. It can also be fused within consecutive kernels.
>
> In practical implementation, the ideal fusion strategy would be to fusing both the reorder and inv-reorder within the QKV linear mapping at the beginning. However, due to the tightly integrated nature of PyTorch’s Linear implementation, it is relatively difficult to hack, and it is also coupled with many other operator-level optimization strategies. Instead, we fuse the reordering process into the relatively "independent" rotary embedding layer, making it easier and more flexible for code integration. As a result, the inverse reordering is performed separately. We verify that the overhead of this operation **remains negligible** (<1%), as shown below:
>
> | InvReorder time (ms) | Attn time (ms)  | Proportion |
> |:--:|:----:|:----:|
> | 0.135    | 20.8          |  0.65%          |
>
> ---
>
> > **Q3:** *Are permute patterns model-specific? minute-level mask generation need to be conducted for different model and input visual information.*
>
> **Response to Q3:**
> - *"Permute patterns model-specific? "* **Yes**, the permute order is closely related to the local aggregation dimension of certain attention head, which corresponds to model weights.
> - *"Minute-level mask generation conducted for different model and input visual information?"* The sparse mask generation needs to be performed for each model, but **not** for each input of visual information. The mask generalizes well across different prompts and random noises, as discussed in detail in the W1 response.
>
> ---
>
> > **Q4:** *How can "reorder-then-compress" benefit large vision language model, the broader impact beyond visual generation of the idea.*
>
> **Response to Q4:** The prefilling stage of autoregressive VLMs share similar computing flow like diffusion models, and the "local aggregation" nature is a general characteristic for all visual generation. Specifically, for VLMs, similar "multi-diagonal" pattern also appears, as presented in the figures for the papers below (We apologize for the indirectness since NeurIPS rebuttal does not support providing figure or links).
> - Right side of Figure 6 "BlindSight: Harnessing Sparsity for Efficient VLMs" (arxiv 2507.09071)
> - Figure 3 of "MMInference: Accelerating Pre-filling for Long-Context VLMs via Modality-Aware Permutation Sparse Attention" (arxiv 2504.16083)
>
> We also apply PAROAttn to **LLada-V[1]**, a diffusion-based vision-language model, to demonstrate its applicability beyond visual generation. We select the MathVerse dataset because it involves high-resolution images, and the image processing accounts for the majority of the computational cost.
>
> | Methods | Mathverse-minivision Score  |
> |:----:|:--:|
> |  Origin    |   28.5         |
> |  Naive (0.75)    |   20.5         |
> |  PAROAttn (0.75)    |   27.8         |
>
> [1] You et. al., "LLaDA-V: Large Language Diffusion Models with Visual Instruction Tuning", arxiv 2505.16933
>
> ---
>
> > **Limitations:**  *static, offline permutation patterns determination and the sparsity metrics*
>
> **Response to Limitations:** To conclude,
> - regarding the *"static offline permute patterns"*, we supplement analysis of design space and generalization (W1 & Q3)
> - regarding the *"sparsity metrics"*, we supplement analysis of hyperparameter robustness (W2), and generalization (W1 & Q3).
>
> ---
>
> We sincerely thank the reviewer for the time and efforts. We hope our responses have addressed your concerns. Please don’t hesitate to let us know if you have any additional questions or feedback on our response. We will do our best to address any concerns you may have :)

---

> > ### Comment · Reviewer_CDie · 2025-08-05
> >
> > Thanks for the detailed and robust responses, which solve all my concerns. I am also happy to see that PAROAttn works for LLada.  I will maintain my current positive rating.

---

> > > ### Author Response · Authors · 2025-08-05
> > > **Gratitude for Feedback and Acknowledgment**
> > >
> > > Thank you for the update. We truly appreciate your acknowledgment of our contributions and response. We are sincerely grateful for your constructive feedback and the time you invested in reviewing our paper.

---

### Official Review · Reviewer_YVHL · 2025-07-02

**Clarity:** 3
**Significance:** 3
**Originality:** 4
**Rating:** 5
**Confidence:** 4

**Summary:**

The authors propose PAROAttention, a lightweight post-training compression scheme for vision diffusion transformers. The key idea is to re-order tokens (PARO) so that the heterogeneous “multi-diagonal” attention maps typical of video/image generators become a single, block-sparse pattern. Once unified, the same block layout is exploited for static sparse masks and block-wise INT8/INT4 quantization, both implemented inside a customized FlashAttention-style CUDA kernel. Experiments on CogVideoX-5B and Flux shows that the proposed method achieves 1.9 – 2.7 × end-to-end speed-ups over FP-dense baselines and up to 10 × when sparsity and quantization are combined.

**Questions:**

Please refer to weaknesses above.

**Ethical Concerns:**

["NO or VERY MINOR ethics concerns only"]

**Final Justification:**

This paper demonstrate solid work on efficient vision generation model. The author address my concerns in the rebuttal. Therefore I would be happy to see this paper is accepted.

**Limitations:**

Yes.

**Quality:**

3

**Strengths And Weaknesses:**

**Strengths**

1. The novelty of the paper is good. PAROAttention proposes to transform the pattern itself through reordering, which leads to more structured and predictable sparsity and quantization domains.

2. The proposed PAROAttention aligns the sparsification and quantization granularity with FlashAttention’s block-wise execution model. The author of the paper provide CUDA compatible design, which is comprehensive.

3. PAROAttention determines the optimal permutation and sparse masks with just 1–2 prompts, which demonstrate easy calibration process of using such design.

4. The overall results are convincing. The gain of the proposed method is significant compared to baselines.

**Weaknesses**

1. All results are reported without variance or error bars. Could the author of the paper show more experiment detail like how many time performed for experiments? And what is the variance for the experimental results.

2. The method uses static sparse masks determined offline using only 1–2 prompts, which is a good thing. However, at inference stage, I assume the prompt distribution should match the one in the calibration stage. This brings concerns when there is distribution shift between calibration and inference. This is also my question: how does the proposed PAROAttention deal with the distribution shift?

---

> ### Author Rebuttal · Authors · 2025-07-31
>
> We sincerely thank reviewer YVHL for the time and efforts, constructive feedback, and insightful suggestions, which are invaluable for improving our paper. We are encouraged by the reviewers’ recognition of the novelty, CUDA-aware comprehensive design, easy practial usage, and solid efficiency improvement. We address each of the raised weaknesses and questions as follows:
>
> > **Weakness 1:** *Variance of the experimental results (missing error bars).*
>
> **Response to W1:** We agree with the reviewer’s suggestion to include error bars to validate the variance in the experimental results. In the main paper, due to the high evaluation cost (video generation across numerous prompts), the experiments were conducted only once. To address this, we have supplemented the experiments by incorporating **additional random seeds to validate the controlled variance** of the evaluation.
>
> | Method (VBench)         | SSIM               | PSNR               |
> |:-----------------:|:------------------:|:------------------:|
> | **PAROAttn (0.5)**  | 0.909 (±0.0107)     | 28.227 (±1.195)    |
>
> As can be seen, although random seeds have some effect on the absolute values of the metrics, likely due to variations in content which causes varying difficulty for preservation, **The overall variance remains relatively low**. It is also worth noting that, despite variations in the metric values, all methods exhibit a similar trend in value changes (i.e., when a seed has a higher metric value, the values for all methods tend to be relatively higher). This suggests that the variance is due to the "relative difficulty" of the seed rather than pure noise.
>
> ---
>
> > **Weakness 2:** *How to deal with the distribution shift from calibration and inference prompts.*
>
> **Response to W2:** We agree with the reviewer that generalization across different conditions (prompts and noise) is a critical issue for static sparse attention approaches. **We do not specifically address the "distribution shift' problem, as our observation shows that 'the attention pattern of the image part remains consistent across different conditions"**. To support this finding, we provide extensive empirical evidence and hypotheses regarding the underlying reasons.
>  - **Empirical observation:** We carefully verify that the attention pattern shows high similarity across different conditions (prompts and noise) through empirical analysis (as discussed in Lines 192-198 of the main paper). We observe an exceptionally high similarity across different prompts (cosine similarity ≥ 0.99). Appendix Figure 8 provides a visualization of the attention patterns, with each row representing a different prompt, and the patterns appear very similar. This indicates that the attention pattern, corresponding to the sparse mask, does not vary significantly across prompts.
>  - **Assumption of underlying reasons:** The prompt primarily affects the text-image attention part, which accounts for only a small portion of the attention (<1%) and is therefore omitted for sparsification. The majority of the attention values, which describe the 'inter-image' aggregation, are more related to the model and are invariant to the input prompts. This phenomenon, where the attention pattern and sparse mask are primarily influenced by model weights rather than input activations, has also been observed in prior literature[1,2].
>  - **Additional Experiments:** To further verify this finding, we also provide results for another prompt set (VBench V2), using the current sparse mask generated from an out-of-domain prompt set (cogvideo example prompts) to demonstrate the model's generalization ability across prompt domains. As can be seen, PAROAttn still demonstrates good generalization ability across prompts and random noises.
>
> | Method (VBench2)        | SSIM              | PSNR               |
> |:----------------:|:----------------:|:------------------:|
> | **PAROAttn (0.5)** | 0.919 (±0.0161)    | 28.442 (±1.054)     |
>
>
> [1] Jiang et. al., MInference 1.0: Accelerating Pre-filling for Long-Context LLMs via Dynamic Sparse Attention.
>
> [2] Xi et. al., Sparse VideoGen: Accelerating Video Diffusion Transformers with Spatial-Temporal Sparsity.
>
> ---
>
> We sincerely thank the reviewer for the time and efforts. We hope our responses have addressed your concerns. Please don’t hesitate to let us know if you have any additional questions or feedback on our response. We will do our best to address any concerns you may have :)

---

> > ### Comment · Reviewer_YVHL · 2025-08-03
> > **Thanks for the rebuttal**
> >
> > I appreciate the efforts that the author put in the rebuttal. My concerns are addressed. I would suggest acceptance of the paper.

---

> > > ### Author Response · Authors · 2025-08-03
> > > **Gratitude for Feedback and Acknowledgment**
> > >
> > > Thank you for the update. We truly appreciate your acknowledgment of our contributions and response. We are sincerely grateful for your constructive feedback and the time you invested in reviewing our paper.

---

### Official Review · Reviewer_wfYy · 2025-07-03

**Clarity:** 3
**Significance:** 3
**Originality:** 3
**Rating:** 4
**Confidence:** 3

**Summary:**

The paper proposes ​PAROAttention, a novel method to accelerate attention mechanisms in visual generation models (e.g., diffusion transformers) via ​pattern-aware token reordering (PARO)​. Key innovations:
​1) Token Reordering: Unifies diverse attention patterns (block-wise, multi-diagonal) into hardware-friendly block-sparse structures by permuting token dimensions ( [F,H,W] → [H,W,F]).
​2) Joint Sparsification & Quantization: Enables ​20-30% density sparsity​ and ​INT8/INT4 quantization​ with near-lossless quality.
3) ​Hardware Efficiency: Achieves ​1.9–2.7× speedup​ via CUDA-optimized kernels and minimal overhead (<1%).

**Questions:**

Question:
) Insufficient Long-Sequence Validation​: PARO primarily evaluates 6-second clips, lacking validation for critical long-sequence scenarios (>10s videos) where 3D full-attention models typically struggle.

Suggestion:
A highly relevant study, "Sparse VideoGen: Accelerating Video Generation with Spatial-Temporal Sparse Attention by 2× with High Fidelity" (SVG Project), demonstrates significant performance improvements through token reordering strategies. While our approach shares the fundamental concept of optimizing local sparsity through token permutation, a systematic comparison would be valuable to elucidate both methodological similarities and key distinctions. Would it be possible to conduct a comprehensive comparison between these works, either through experimental validation or theoretical analysis?

**Ethical Concerns:**

["NO or VERY MINOR ethics concerns only"]

**Limitations:**

In video generation, attention mechanisms typically manifest in two distinct forms: ​spatial attention, which focuses on local pixel relationships within individual frames, and ​temporal attention, which tracks object movements across frames. The paper fails to adequately differentiate between these fundamentally different attention patterns, potentially compromising the efficiency of its proposed approach.

**Quality:**

3

**Strengths And Weaknesses:**

Strengths:
1) ​The primary contribution is indeed ​token reordering (PARO)​, a straightforward yet effective engineering approach to optimize attention computation.
2) The proposed algorithm outperforms baselines (DiTFastAttn, SpargeAttn) by ​8–12% accuracy​ at 50% sparsity.

Weaknesses:​
1) The proposed algorithm lacks of theoretical depth. There is no analysis of why certain permutations work best or bounds on reordering error.
2) Table 3 shows PARO's performance degrades without timestep sharing, indicating weaker adaptation to temporal dynamics.

---

> ### Author Rebuttal · Authors · 2025-07-31
>
> We sincerely thank reviewer wfYy for the time and efforts, constructive feedback, and insightful suggestions, which are invaluable for improving our paper. We are encouraged by the reviewers’ recognition of the novelty of token reordering (PARO) and the solid improvement of the performance–efficiency trade-off. We address each of the raised weaknesses and questions as follows:
>
> > **Weakness 1:** *lack of theoretic analysis of why certain permutations work best or bounds on reordering error.*
>
> **Response to W1:** We agree with the reviewer that choosing proper reordering strategy is a critical issue. The scope of the current paper is to introduce the motivation for reorganizing attention patterns, and propose a feasible token reordering strategy to realize this idea. **The current reordering design is relatively simple and serves as a proof of concept to validate its effectiveness**, while leaving room for future improvements to further enhance its potential.
> - **The current PARO plan is empirically verified effective despite simplicty:** The current PARO design is guided by the strong prior of the local aggregation nature inherent in visual feature extraction, which is supported by extensive empirical observations. To reduce the complexity of optimization, we adopt an extremely small search space consisting of only 6 possible permutations. In this setting, identifying *"which permutations work best"* becomes tractable through simple enumeration. Despite the simplicity of the design and the limited search space, PARO demonstrates strong pattern reorganization capabilities, effectively generating unified block-wise patterns in most cases, as illustrated in Appendix Figures 5 and 8.
> - **The design of PAROAttn is compatible with more advanced optimization strategies for token reordering**, allowing the pursuit of "lower bounds on reordering error". Beyond the PARO reorder, PAROAttn also provides extensive analysis of sparsification design choices and finally adopts a static approach that determines both the sparsity and reordering strategies offline. This design enables more flexible and sophisticated optimization, as it avoids incurring additional online overhead. To further demonstrate this potential, we supplement our experimental results with an extension that integrates the existing PAROAttn framework with balanced K-Means clustering.
>
> |      Method (CogVideo)      | PSNR (videos) | SSIM (videos) |
> |:----:|:----:|:---:|
> | PARO      |     19.61     |     0.753     |
> | PARO+Cluster     |     20.37     |     0.790     |
>
> As can be seen, combining balanced clustering with the current PAROAttn yields only a slight performance improvement. This is consistent with the results in Q2 below (PAROAttn achieves similar performance with the clustering-based SVG-V2). The underlying reason for this phenomenon is that the clustering objective focuses on grouping similar values together (similar to the "local uniform" in the main paper's Fig. 7), rather than achieving a complete "sparse block". **These results validate both the effectiveness of the current PARO strategy and its compatibility with more advanced optimization techniques.**
>
> ---
>
> > **Weakness 2:** *Table 3 shows PARO performs worse without timestep sharing, suggesting limited temporal adaptability.*
>
> **Response to W2:** We would like to clarify that the purpose of this ablation study is to evaluate the effect of removing timestep sharing by applying a distinct sparse mask for each timestep. It is expected to yield improved results at the cost of increased resource consumption, the observation of no improvement, or even degradation in some metrics suggests that timestep-wise masking does not offer additional performance benefits. This, in turn, **supports the effectiveness and rationality of our proposed timestep-sharing strategy**. In PAROAttention, ***"temporal adaptability"*** can be ensured through timestep-wise sparse masks, and the associated memory overhead can be mitigated via prefetching, as discussed in Lines 190–201 of Sec. 4.2. We will carefully revise the manuscript to further improve clarity and prevent potential misinterpretation.
>
> ---
>
> > **Q1:** *PARO lacks validation on longer sequences (>10s)*
>
> **Response to Q1:** We agree with the reviewer that long-sequence scenarios pose greater efficiency challenges. **Experimental results and analysis for 10s generation of Wan 2.1 model are provided in Appendix Sec. 1, Table 1, and Figure 1.** Due to space limits, for clarity, we present and discuss part of these results **in Q2 below**. We thank the reviewer for this valuable suggestion and will carefully reorganize and integrate the content into the main text in future revision.
>
> ---
>
> > **Q2:** *Suggestion of comparing with SVG project*
>
> **Response to Q2:** We agree that the SVG project is closely related to our work. It consists of two research papers. We conduct a detailed comparison with the initial ICML submission (arXiv:2502.01776) as an important baseline. The updated SVG-V2 paper (arXiv:2505.18875), which introduces token reordering strategies, was published on arXiv on May 24—after the NeurIPS submission deadline of May 15. Therefore, it was not included in our original submission. We now provide a detailed discussion of SVG-V2 as follows:
>
> - **Methodology Comparison:** Both SVG-V2 and PAROAttn recognize the challenge posed by dispersed attention values and propose token reordering as a solution. However, compared to PAROAttn, SVG-V2 adopts a more advanced reordering strategy and dynamic sparsification, aiming for higher potential in algorithmic performance, with the cost of relatively compromised efficiency gains.
>      - **Distinction 1: How to determine the optimal reordering strategy.** As discussed in W1, PAROAttn adopts a relatively simple reordering scheme as a proof of concept, while remaining compatible with more advanced optimization strategies—such as the K-Means-based approach used in SVG-V2.
>          - As shown in the experiments in W1, PARO demonstrates strong effectiveness despite its simplicity, achieving performance comparable to SVG-V2. Moreover, it can be further improved when combined with more sophisticated reordering strategies.
>      - **Distinction 2: Dynamic vs. Static Sparse Approach.** PAROAttn conducts an extensive analysis of design choices (Sections 3 and 4) and ultimately adopts a static sparse approach, whereas SVG-V2 employs a dynamic approach in which both the reordering strategy and sparse mask are generated online.
>           - As discussed in W1, the static approach enables more flexible and sophisticated optimization by avoiding additional online overhead. For instance, it supports time-consuming techniques such as balanced K-Means, which ensures equal token counts across clusters (while SVG-V2 adopts normal kmeans due to online overhead). This design helps mitigate the need for specialized kernels to handle diverse block shapes and alleviates performance degradation caused by load imbalance.
>      - **Distinction 3: Discussion on effects of Quantization.** Additionally, PAROAttn provides discussion and experiments regarding how token reorder benefits quantization.
>
> - **Experimental Validation:** Due to the unavailability of the customized CUDA kernel code for SVG-V2, we focus our comparison on algorithmic performance and provide a discussion of hardware efficiency.
>     - **Algorithm Performance:** We report results on 10-second long-sequence generation using the Wan v2.1 model (as discussed in Q1). As shown, PAROAttn achieves comparable performance to SVG-V2 despite employing a simpler reordering strategy.
>     - **Hardware efficiency:** Compared to PAROAttn, SVG-V2 introduces additional overheads, including the following:
>          - **(1) Overhead from K-Means:** The cost of K-Means clustering is amortized across timesteps and accounts for about 1% of the attention cost. However, this percentage increases and becomes non-negligible for shorter sequences or higher sparsity rates.
>          - **(2) Increased Reordering Overhead:** Unlike PAROAttn, SVG-V2 determines the reordering strategy online, makes it difficult for kernel fusion of the reorder. The standalone reorder for Q, K, V, O can introduce an additional 5–10% overhead of attention cost.
>          - **(3) Load Imbalance:** K-Means clustering introduces variability in the number of tokens assigned to each cluster, requiring customized FlashAttention kernels to support arbitrary block sizes. It also causes load imbalance issues, notably reducing hardware efficiency.
>
> |      Method (Wan)       | PSNR (videos) | SSIM (videos) |
> |:---:|:----:|:-------------:|
> |   SVG-V2     |     25.81     |     0.854     |
> | PAROAttn     |     25.45     |     0.887     |
>
> ---
>
> > **Limitation:** *Missing differentiation between spatial vs. temporal attention may hinder method efficiency.*
>
> **Response to Limitation:** We would like to clarify that PAROAttn **adaptively differentiates between spatial and temporal heads when determining the optimal permutation order**. Specifically, it selects the appropriate local aggregation dimension for each attention head. For example, in a permutation order such as [W, H, F], the frame dimension (F) remains contiguous in memory. This order will be selected when the permuted attention map exhibits a block-wise pattern, indicating that the current head is a temporal head; a similar rationale applies to spatial attention. Consequently, the pattern-aware reordering process effectively unifies both spatial and temporal attention into block-wise pattern. It preserves efficiency without requiring explicit differentiation between the two types of attention.
>
> ---
>
> We sincerely thank the reviewer for the time and efforts. We hope our responses have addressed your concerns. Please don’t hesitate to let us know if you have any additional questions or feedback on our response. We will do our best to address any concerns you may have :)

---

### Note · Authors · 2025-08-12

We sincerely thank reviewers, AC/SACs/PCs' time and efforts. We are encouraged to receive positive feedback from all reviewers prior to the rebuttal stage, and we appreciate that our response addressed the reviewers’ concerns and reach a consesus recommendation for acceptance.

We introduce **PAROAttention**, providing a systematic analysis of the challenges inherent to attention sparsification and quantization in diverse and irregular visual attention pattern, and propose a novel token-reordering strategy that simplifies and enhances both techniques. We further develop an efficient CUDA implementation to validate the method’s practical hardware acceleration.

In the initial reviews, we were encouraged by reviewers’ recognition of:
1. **Novelty, simplicity, clear motivation, and effectiveness** of the key token reorder design (Reviewers: wfYy, YVHL, CDie, A1jm).
2. **Solid performance gains**: preserving generation quality while delivering significant hardware acceleration over baseline methods (Reviewers: wfYy, YVHL, CDie, A1jm).
3. **Flexibility and practicality:** It requires only minutes of adaptation, and complete CUDA code is provided for practical use (Reviewers: wfYy, CDie).
4. **Clarity of writing and presentation** (Reviewer: CDie).

During the author's response, we address the following concerns:

1. **The sufficiency of the PARO reorder strategy.** (Review wfYy, CDie, A1jm) We clarify that current design is a simple proof-of-concept, and add experiment to demonstrate the compatibility.
2. **The generalization of static approach.** (Review YVHL, CDie)  We validate the observation through both visualization, and experiments of generalizing to novel unseen dataset VBench-V2.
3. **Applicablilty to broader models.** (Review CDie, A1jm) We demonstrate it is applicable to both prefilling stage of autoregressive VLM, and diffusion VLMs (e.g., LLaDaV).

We sincerely thank the reviewers for the suggestions, which have really helped us further refine and strengthen the paper. We hope that by exploring the intrinsic “local aggregation” property of visual features, PAROAttn could **potentially inspire further research on the novel direction of "token reorder", and benefit a broader range of fields.

We want to additionally thank the AC/SACs/PCs' for their effort and dedication in fostering constructive communication and a rigorous peer-review process, and for upholding the quality of reviews despite the challenges of a large-scale submission.

---

### Decision · Program_Chairs · 2025-09-17

**Decision:**

Accept (poster)

**Comment:**

The final ratings for this paper are unanimously positive (two "Accept" and two "Borderline Accept").

The paper introduces PAROAttention, a framework for accelerating visual generation models. The core idea of reorganizing attention patterns via a simple token-reordering strategy and speedups with minimal quality loss are strong contribution of this paper.

Initial reviews raised valid questions about the generalizability of this static approach. In a rebuttal, the authors addressed these concerns by demonstrating the method's robustness on an unseen dataset (VBench-V2) and applying it to a different architecture, a diffusion-based VLM (LLaDa-V).

The AC agrees that this is a practical contribution and recommends the paper for acceptance.